# CaliDist: Calibrating Large Language Models via Behavioral Robustness to Distraction

## Abstract

For Large Language Models (LLMs) to be trusted in high-stakes applications, it is paramount that their confidence scores are well-calibrated. However, existing calibration methods often overlook a critical dimension of trustworthiness: a model's behavioral robustness to irrelevant or misleading information. In this paper, we argue that a model's true confidence should reflect its stability under cognitive pressure. We introduce CaliDist, a novel, post-hoc calibration framework that directly measures and penalizes a model's susceptibility to distraction. CaliDist quantifies how an LLM's predictions and certainty change when its input prompt is perturbed with semantic *distractors*. This instability signal is then used to adaptively scale the model's initial confidence score. Our extensive experiments on seven Natural Language Understanding (NLU) classification benchmarks using six distinct LLMs show that CaliDist consistently achieves lower Expected Calibration Error (ECE) than several baselines. Remarkably, our method reduces the ECE from 19% to 11% on average—a relative improvement of 47%—demonstrating that behavioral stability is a powerful and practical signal for calibration.

## 1 Introduction

Large Language Models (LLMs) have demonstrated remarkable capabilities across a vast spectrum of complex tasks, leading to their rapid integration into high-stakes domains such as medical diagnostics, legal analysis, and financial advising (Thirunavukarasu et al., 2023). In these critical applications, the correctness of a model's output is paramount, but equally important is the reliability of its self-assessed confidence. A well-calibrated model expresses confidence that accurately reflects the true likelihood of its correctness and is essential for building trustworthy systems, enabling safe adoption into such critical systems, and knowing when to defer to a human expert Sun et al. (2024). However, modern LLMs, particularly those fine-tuned with Reinforcement Learning from Human Feedback (RLHF), are often severely miscalibrated, typically exhibiting a strong tendency towards overconfidence (Achiam et al., 2023; Tian et al., 2023).

Existing research on model calibration has largely followed two main paradigms. The first, inherited from traditional deep learning, involves post-hoc statistical adjustments, such as temperature scaling (Guo et al., 2017). While effective, these methods are limited by their requirement for direct access to model logits, rendering them inapplicable to many proprietary LLMs. The second paradigm, developed for the generative nature of LLMs, estimates confidence by measuring response consistency, most notably through self-consistency (Wang et al., 2023). These methods cleverly probe the model's internal certainty but often incur significant computational costs. More importantly, they measure consistency against the model's own internal stochasticity, not against external challenges.

This leaves a critical dimension of trustworthiness unexplored: a model's behavioral robustness. Recent studies have shown that LLMs are notoriously brittle, often changing their predictions when presented with irrelevant but plausible "distractor" information (Shi et al., 2023). This brittleness mirrors well-documented phenomena in human cognitive psychology, allowing us to frame our approach. Specifically, we can draw two parallels, each corresponding to a distinct, measurable behavior: First, a model's tendency to alter its original, correct prediction after being exposed to misleading information is a computational parallel to the *Misinformation Effect* (Loftus & Palmer, 1974), where human memory of an event can be fundamentally altered by misleading post-event information. Here, the model's initial prediction acts as its "memory" of an answer, and a seman-

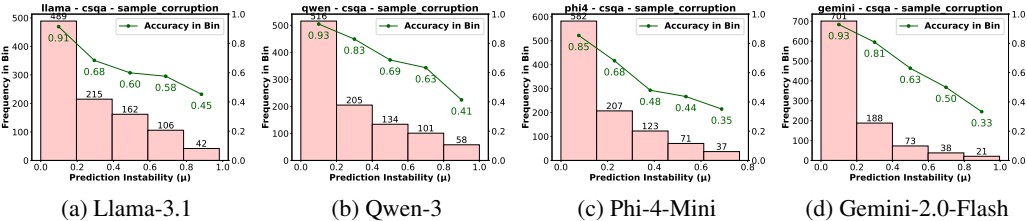

Figure 1: Negative correlation between prediction accuracy and prediction instability. Accuracy drops with an increase in Prediction Instability $\mu$. Samples on which the models demonstrate higher $\mu$ tend to have lower average accuracy. Distractor used: *Sample-corruption* style.

tic distractor serves as post-event information that can erroneously corrupt that memory. A model highly susceptible to this effect is inherently less reliable, as it reveals a fundamental flaw in its reasoning process: it indicates that the model's conclusions are not based on a robust, internal understanding of the problem, but are instead heavily influenced by superficial cues in the prompt. Second, a model that remains highly confident while being easily distracted exhibits a behavior akin to the *Dunning-Kruger Effect*, where low competence on a task is paired with an inflated overestimation of ability (Kruger & Dunning, 1999). In this parallel, the model's "incompetence" is its inability to resist distraction, and its "overestimation" is its failure to reduce its confidence accordingly.

To operationalize these parallels, we introduce CALIDIST, a novel, post-hoc framework that quantifies and aggregates these behavioral signals for calibration. CALIDIST systematically perturbs an input prompt with targeted distractors and uses the combined instability signal from both prediction changes and confidence shifts to adaptively scale the model's original confidence score. CALIDIST measures the Misinformation Effect parallel with Prediction Instability ($\mu$), which quantifies how frequently a model's prediction changes when perturbed. CALIDIST also captures the Dunning-Kruger parallel by observing this instability in conjunction with the model's Confidence Stability ($\delta$), or its change in certainty.

Crucially, we validate that these behavioral metrics are not just quirks but are directly linked to correctness. As visualized in Figure 1, we observe a strong and consistent negative correlation between Prediction Instability ($\mu$) and the model's accuracy on the original, unmodified prompts. We tested three open-source and one proprietary model and observed that samples exhibiting low instability (e.g., $\mu \leq 0.2$) have a very high average accuracy (often above 80-90%), whereas accuracy drops sharply as instability increases. This finding provides the foundational evidence for our work: susceptibility to distraction is a powerful, measurable proxy for a model's likelihood of error. Because our approach can operate on log-probabilities or verbalized confidences (Tian et al., 2023; Xiong et al., 2024), it is broadly applicable to both white-box and black-box LLMs.

The main contributions of this paper are: (1) We introduce behavioral robustness to distraction as a new and critical dimension for LLM calibration, grounding it in established principles from cognitive psychology, such as the Misinformation Effect and the Dunning-Kruger Effect. We empirically demonstrate its strong correlation with prediction accuracy; (2) We propose CALIDIST, a novel, post-hoc calibration approach that quantifies a model's stability against semantic distractors to adaptively adjust per-sample confidence scores; (3) We demonstrate the versatility of CALIDIST, showing its effectiveness for both white-box and black-box LLMs; (4) We provide extensive empirical evidence showing that our method significantly reduces Expected Calibration Error (ECE) and Brier Score (BS) across multiple datasets and LLMs.

## 2 RELATED WORK

Our work is situated at the intersection of three key areas: (1) post-hoc calibration methods, (2) consistency-based confidence estimation, and (3) the study of LLM robustness to adversarial inputs.

**Post-Hoc and Black-Box Model Calibration.** Post-hoc calibration remaps a model's output probabilities without altering its weights. The most common method, Temperature Scaling (TS) (Guo et al., 2017), divides the logits by a learnable scalar $T$, but like other foundational methods such

as Platt Scaling (Platt et al., 1999) and Isotonic Regression (Zadrozny & Elkan, 2002), it requires access to model logits. Recent works have extended these ideas to LLMs, for instance, by learning task-specific temperatures (Shen et al., 2024) or adapting TS for semantic-level confidence (Lamb et al., 2025). The primary limitation of these approaches is their inapplicability to black-box APIs. The challenge of calibrating black-box models has spurred research into methods that do not require logit access. One prominent direction is verbalized confidence, where the model is prompted to state its certainty directly (Xiong et al., 2024). While often miscalibrated, these scores can provide a stronger signal than the conditional probabilities of RLHF-tuned models (Tian et al., 2023; Xiong et al., 2024). Other approaches involve training auxiliary models to predict correctness (Ulmer et al., 2024; Pedapati et al., 2024), or using conformal prediction (Azaria & Mitchell, 2023). Our method is distinct as it requires no external model, deriving its signal directly from the target model's behavior. CALIDIST additionally circumvents the problem of calibrating black-box models by being fully compatible with log-probabilities or verbalized confidence, effectively acting as a behavioral proxy for TS that achieves a similar confidence-scaling effect without needing logit access.

**Consistency-Based Confidence Estimation.** A separate family of methods estimates confidence by measuring the consistency of a model's outputs across multiple forward passes (Wang et al., 2023; Manakul et al., 2023; Xiong et al., 2024; Wightman et al., 2023). These techniques, which can measure semantic similarity (Lamb et al., 2025) or aggregate votes from paraphrased prompts (Kadavath et al., 2022), often incur substantial computational overhead, typically requiring 10-40 passes for a stable signal (Manakul et al., 2023). Our method shares the multi-pass approach but is fundamentally different: instead of measuring consistency across stochastic samples of an identical prompt, we measure stability against a set of deterministic, adversarial distractor prompts, allowing us to achieve a robust signal with significantly fewer forward passes.

**LLM Robustness to Adversarial Context.** A parallel stream of research has established that LLMs are brittle and easily swayed by irrelevant context or distractors (Shi et al., 2023; Chen et al., 2024; Mozes et al., 2023; Huang et al., 2025). Xiong et al. (2024) employ induced consistency to sample multiple responses. While existing work uses these failures to demonstrate LLM limitations, our work is the first to formally bridge the concept of distraction robustness with the task of confidence calibration, leveraging this behavioral signal as a core component of our framework.

## 3 METHODOLOGY

In this section, we elaborate on the working principle of the CALIDIST framework. Our proposed framework calibrates the confidence of an LLM by evaluating its behavioral robustness. Instead of relying solely on a model's initial, static confidence score, our method leverages the stability (or lack thereof) of its predictions and certainty when presented with "distractor" information. The core intuition is that a reliable model should remain stable in both its conclusion and its associated confidence, even under cognitive load. We show the operational flow of CALIDIST in Figure 2. The algorithm is presented in Appendix 8.1.

### 3.1 FORMALISM AND NOTATION

Let $M$ be the model, $\pi_o$ be the original input prompt, $\mathbf{x}$ be the input, $y$ be the ground truth, and $\hat{y}$ be the model's initial prediction, for which it produces an initial confidence score $p = P(\hat{y}|\pi_o(\mathbf{x}))$. This confidence can be derived from either the logit-based probability of the output token (for white-box models) or from the log-probability and verbalized confidence (for both white-box and black-box models). We define a set of $k$ distractors $D = \{d_1, d_2, \ldots, d_k\}$. Each distractor $d_j$ is a piece of information designed to be semantically related to the task but logically irrelevant or contradictory to the initially predicted reasoning path for the input $\mathbf{x}$. For each distractor $d_j$, we construct a new, distracted prompt $\pi_{d_j} \leftarrow \pi_o \oplus d_j$, where $\oplus$ denotes the concatenation or integration of the distractor into the original prompt. The model's forward pass on this new prompt yields a new prediction $\tilde{y}_{d_j}$ with a corresponding confidence score $p'_j = P(\tilde{y}_{d_j}|\pi_{d_j}(\mathbf{x}))$.

### 3.2 DISTRACTOR STYLES

We design three distinct styles of distractors to induce different types of perturbations in the responses of LLMs: *Assertion-style*, *Probe-style*, and *Sample-Corruption-style* distractors.

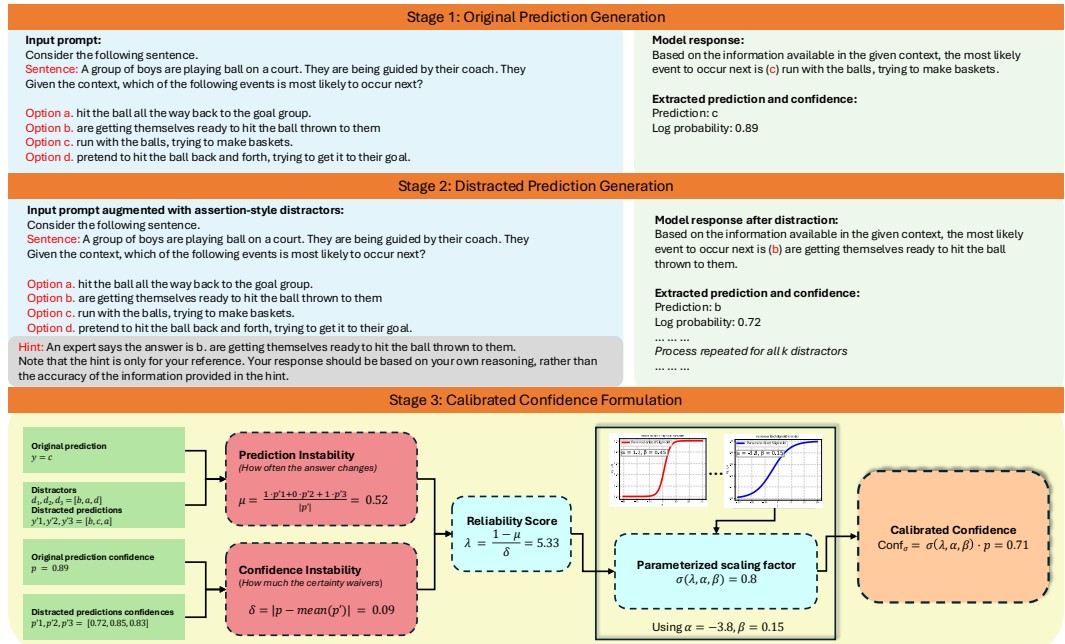

Figure 2: Illustration of the CALIDIST framework. Note: This example shows one of $k = 3$ *assertion-style* distractor prompts. The process in Stage 2 is repeated for each distractor to gather a set of distracted predictions and confidences. For brevity, we show only one distractor prompt and response generation in Stage 2. For our experiments with *assertion-style* distractors, we report our findings with $k = (c-1) \cdot m$ distractors, where $m = 2$ denotes the number of distractors generated per class and $c$ is the total number of classes.

**Assertion-style Distractors.** Assertion-style distractors are authoritative assertions appended with the original prompt to try to deviate a model's initial response. For example, an assertion-style distractor may state, "*Wikipedia claims the answer is . . ..*", followed by a misdirecting label.

**Probe-style Distractors.** These distractors are designed to encourage the model to consider alternative answers, thereby inducing uncertainty and checking for deviation from its original response. An example of a probe-style distractor is, "*Do you think the answer could be . . .?*"

**Sample-Corruption-style Distractors.** This style introduces a disruptive paradigm by corrupting the samples themselves with misleading cues. For instance, in the MNLI benchmark, if the premise is *"She is known to many as the Queen of Rockabilly"* and the hypothesis is *"She is known to many as the Queen of Rockabilly or the First Lady of Rockabilly"*, the hypothesis can be corrupted by adding *"This sentence contradicts the premise."* as a distractor. The primary goal of this distractor style is to observe any deviations in the model's responses caused by distractors embedded in the sample and to use these deviations as calibration signals.

We show the detailed design and prompt templates for these distractor styles in Appendix 8.4 and 8.5.

## 3.3 DISTRACTION-BASED CALIBRATION FRAMEWORK

Our framework follows a systematic, multi-step process for each input sample to derive a final, calibrated confidence score.

**Elicit Initial Prediction.** First, we use the original prompt $\pi_o$ on sample $\mathbf{x}$ to obtain the baseline prediction $\hat{y}$ and its associated confidence $P(\hat{y}|\mathbf{x})$.

**Generate Distractors and Elicit Distracted Predictions.** For a given classification task with $c$ possible labels, we generate $k = (c-1) \times m$ distractors, where $m$ is a hyper-parameter that denotes the number of distractors to generate per class. Each distractor is constructed based on one of the originally unselected labels, creating a set of plausible alternative contexts. For instance, in the

MNLI task with labels {Entailment, Contradiction, Neutral}, if the original prediction $\hat{y}$ is "Contradiction", we generate distractors related to "Entailment" and "Neutral". This creates a structured and principled set of challenges for the model. We use the $k$ distracted prompts $\pi_{d_1}, \cdots, \pi_{d_k}$ to obtain the prediction $\tilde{y}_{d_j}$ and its confidence $P(\tilde{y}_{d_j}|\pi_{d_j}(\mathbf{x}))$ for all $j = 1, \cdots, k$.

**Quantify Prediction and Confidence Instability.** Using the set of responses from the distracted prompts, we calculate two core metrics of instability:

- **Prediction Instability ($\mu$):** Prediction Instability measures the frequency with which the model changed its prediction from the original answer $\hat{y}$. It is calculated as the weighted confidence scores of the fraction of distracted predictions $\tilde{y}_{d_j}$ that do not match $\hat{y}$:

$$\mu \leftarrow \frac{1}{k} \sum_{j=1}^{k} \mathbb{I}(\tilde{y}_{d_j} \neq \hat{y}) \cdot P(\tilde{y}_{d_j}|\pi_{d_j}(\mathbf{x})) \tag{1}$$

  A high $\mu$ indicates the model is easily swayed by irrelevant information.

- **Confidence Instability ($\delta$):** This metric measures the magnitude of the shift in the model's average confidence level between its original prediction and its predictions under distraction. It is defined as:

$$\delta \leftarrow \left| P(\hat{y}|\pi_o(\mathbf{x})) - \frac{1}{k} \sum_{j=1}^{k} P(\tilde{y}_{d_j}|\pi_{d_j}(\mathbf{x})) \right| \tag{2}$$

  A large $\delta$ signifies that the model's certainty is volatile and unreliable.

**Calculate Reliability Score ($\lambda$).** We combine the two instability metrics into a single reliability score, $\lambda$. This score is designed to be high when both $\mu$ and $\delta$ are low, indicating a robust and stable model.

$$\lambda = \frac{1 - \mu}{\delta + \epsilon} \tag{3}$$

where $\epsilon$ is a small constant (e.g., $1e - 10$) to prevent division by zero.

**Final Confidence Calibration.** Finally, the reliability score $\lambda$ is passed through a parameterized sigmoid function, $\sigma$, to produce a scaling factor between 0 and 1. This factor is then multiplied by the original confidence to yield the final calibrated confidence, $\text{Conf}_\sigma$.

$$\text{Conf}_\sigma = \sigma(\lambda, \alpha, \beta) \cdot P(y|x), \text{ where } \sigma(\lambda, \alpha, \beta) = \frac{1}{1 + e^{-\beta \times (\lambda - \alpha)}} \tag{4}$$

To illustrate the intuitive logic of our framework, we outline its behavior across four key scenarios in Appendix 8.2.

## 4 EXPERIMENT SETUP

**Datasets.** To evaluate the effectiveness of our proposed CALIDIST framework, we conduct experiments across seven diverse and challenging benchmarks spanning multiple Natural Language Understanding (NLU) tasks. For Natural Language Inference (NLI), we use MNLI (Williams et al., 2018), a large-scale, multi-genre corpus, and MSciNLI (Sadat & Caragea, 2024), which specifically tests inference over scientific texts. To assess paraphrase identification, we include the Twitter PPDB (Lan et al., 2017), a dataset of paraphrase pairs drawn from social media. For Commonsense Reasoning, we employ HellaSwag (Zellers et al., 2019), a sentence completion task that requires predictive reasoning, and Commonsense QA (Talmor et al., 2019) a multiple-choice question-answering benchmark. Finally, to test performance on more complex reasoning and knowledge-intensive tasks, we use Yahoo Answers (Zhang et al., 2015), a large-scale topic classification dataset, and AQuA-RAT ((Ling et al., 2017), a collection of algebraic word problems that require step-by-step rationales. For each dataset, we procure 1000 samples from their respective test sets and a separate set of 200 samples as the held-out validation set (when explicit validation sets are unavailable). This selection of datasets enables us to evaluate our method's performance across a diverse range of domains and reasoning types.

**Models.** Our experiments are conducted on a diverse suite of six state-of-the-art language models to ensure that our findings are broadly applicable. For open-source models, we use four prominent LLMs: Llama-3.1 8B Instruct (Grattafiori et al., 2024), Qwen3 8B (Yang et al., 2025), Phi-4-mini Instruct (Abouelenin et al., 2025), and Gemma-3 4B Instruct (Kamath et al., 2025). These models were selected for their strong performance and varying architectural designs. To validate the effectiveness of our framework in black-box scenarios where logit access is unavailable, we also evaluate two leading proprietary models accessed via their APIs: GPT-4o-Mini (OpenAI, 2024) and Gemini 2.0 Flash (Google, 2024). These models expose token-wise log-probabilities, enabling our evaluation protocol to be applied to a set of different types of confidence values.

**Baselines.** We evaluate the performance of our CALIDIST framework against several well-established baselines to contextualize its effectiveness. Given that our method uses multiple forward passes to assess reliability, our primary comparison is against state-of-the-art consistency-based approaches. We implement three variants: Self-Consistency (Wang et al., 2023), which relies on a simple majority vote over stochastic samples; Entropy-based Consistency, which uses the entropy of the output distribution as a confidence measure; and First-Second-Distance Consistency (FSD), which measures the probability gap between the top two most frequent predictions (Lyu et al., 2025). For all consistency methods, we use a standard of 15 forward passes per sample. Additionally, for white-box models where logits are accessible, we compare our approach with TS (Guo et al., 2017). Since our method can be viewed as a behavioral proxy for TS in black-box settings, this comparison serves as an important point of reference. Finally, we include the uncalibrated Vanilla Confidence (i.e., the model's raw output probability) and verbalized confidence (i.e., the verbalized confidence generated as a response) as baselines to quantify the absolute improvement gained by each calibration method. Please refer to Appendix 8.2 for a detailed description of each baseline.

**Our Method: CALIDIST Variants.** We experiment with two different applications of CaliDist, based on the distractor style and the confidence value used. For example, CaliDist (As.) uses the default probability-based confidence with assertion-style distractor, while the verbalized variant is denoted as CaliDist$_{verbalized}$ (As.) We provide additional details about our approach in Appendix 8.6.

**Evaluation Metrics.** To evaluate the calibration of our models, we use two standard metrics – Expected Calibration Error (ECE) and Brier Score (BS). Details about these metrics can be found in Appendix 8.7.

**Hyperparameter Tuning.** The parameters $\alpha$ and $\beta$ in the final sigmoid function are critical for tuning the calibration behavior. These are not fixed values but are determined empirically for each task. We use a held-out validation set to perform a grid search over a predefined range of values for $\alpha$ and $\beta$. The optimal combination is selected as the one that minimizes ECE on this validation set. This ensures that the penalty function is well-suited to the specific task and model being evaluated.

Additional implementation details can be found in Appendix 8.8.

Table 1: Comparison of CALIDIST with four baselines across seven datasets and two open-source LLMs. Confidence used for all baselines except for Consistency, Entropy, and FSD are logit-based confidence scores. Metrics are given by $\times 10^2$. The best-performing values are in **bold**.

| LLM | Metric | MSciNLI | | MNLI | | PPDB | | Yahoo | | HellaSwag | | CSQA | | AQUA | |
|---|---|---|---|---|---|---|---|---|---|---|---|---|---|---|---|
| | | ECE↓ | BS↓ | ECE↓ | BS↓ | ECE↓ | BS↓ | ECE↓ | BS↓ | ECE↓ | BS↓ | ECE↓ | BS↓ | ECE↓ | BS↓ |
| **LLAMA-3.1 8B** | Temperature Scaling | 12.10 | 22.38 | 5.83 | 19.27 | 5.12 | 18.98 | 12.64 | 17.72 | **1.98** | 16.87 | 5.57 | 14.73 | 8.71 | 17.11 |
| | Vanilla | 25.76 | 28.30 | 2.81 | 19.01 | 13.50 | 20.87 | 19.74 | 21.18 | 8.66 | 17.65 | 12.59 | 16.57 | 23.18 | 22.82 |
| | Consistency | 34.30 | 34.84 | 11.29 | 21.03 | 19.35 | 23.34 | 25.01 | 24.82 | 17.82 | 20.92 | 17.87 | 19.41 | **4.88** | **14.51** |
| | Entropy | 28.05 | 30.75 | 32.14 | 32.44 | 26.68 | 27.88 | 19.89 | 21.25 | 18.83 | 21.59 | 18.37 | 19.71 | 33.22 | 28.20 |
| | FSD | 29.09 | 29.09 | 17.99 | 24.51 | 23.26 | 24.78 | 21.04 | 22.65 | 16.03 | 20.41 | 15.95 | 18.89 | 20.07 | 19.63 |
| | CaliDist (As.) | **4.71** | 22.24 | 2.80 | 19.01 | **2.46** | 18.02 | 11.55 | 18.28 | 3.26 | **16.85** | 6.87 | 15.22 | 14.40 | 17.63 |
| | CaliDist (Pr.) | 6.54 | 22.67 | 2.77 | **19.00** | 2.78 | **17.34** | **10.50** | 18.07 | 4.57 | 17.05 | 5.94 | **15.07** | 14.91 | 18.19 |
| | CaliDist (Sa.) | 6.14 | **21.77** | **2.68** | 19.01 | 6.15 | 17.74 | 11.53 | **17.86** | 3.05 | 17.31 | **2.54** | 15.16 | 15.98 | 19.17 |
| **QWEN-3 8B** | Temperature Scaling | 35.46 | 35.72 | 27.16 | 28.35 | 38.80 | 39.05 | 28.45 | 28.36 | 15.95 | 16.92 | 14.83 | 15.86 | 8.28 | 10.63 |
| | Vanilla | 38.60 | 38.60 | 29.37 | 29.87 | 39.46 | 39.66 | 29.42 | 29.38 | 18.05 | 18.42 | 16.39 | 16.91 | 12.54 | 12.70 |
| | Consistency | 40.14 | 40.14 | 30.25 | 30.54 | 39.47 | 39.46 | 29.86 | 29.89 | 18.82 | 18.76 | 14.87 | 30.37 | 9.15 | 9.63 |
| | Entropy | 38.23 | 39.07 | 30.65 | 30.89 | 38.96 | 38.87 | 29.05 | 29.21 | 17.12 | 17.83 | 14.86 | 30.40 | 8.90 | 10.86 |
| | FSD | 38.87 | 39.38 | 30.58 | 30.74 | 39.10 | 39.14 | 29.52 | 29.57 | 17.86 | 18.15 | 17.17 | 17.36 | 7.37 | **9.48** |
| | CaliDist (As.) | 16.05 | 26.08 | 18.35 | 25.78 | 33.15 | 35.74 | 22.85 | 25.67 | **6.34** | **14.54** | 12.51 | 14.82 | 8.51 | 9.63 |
| | CaliDist (Pr.) | 29.73 | 34.90 | 22.90 | 27.70 | 39.12 | 39.54 | **19.25** | **23.36** | 9.92 | 15.08 | 12.10 | 14.52 | **7.21** | 9.54 |
| | CaliDist (Sa.) | **8.04** | **23.67** | 17.92 | 25.11 | 25.15 | 30.36 | 24.63 | 26.42 | 11.37 | 15.65 | **8.91** | **13.68** | 9.04 | 10.44 |

Table 2: Comparison of CALIDIST with the verbalized confidence method using two open-source LLMs. Metrics are given by $\times 10^2$. The best-performing values are in **bold**.

| LLM | Metric | MNLI | | PPDB | | Yahoo | | HellaSwag | | AQUA | |
|---|---|---|---|---|---|---|---|---|---|---|---|
| | | ECE↓ | BS↓ | ECE↓ | BS↓ | ECE↓ | BS↓ | ECE↓ | BS↓ | ECE↓ | BS↓ |
| LLAMA-3.1 | Verbalized | 26.40 | 28.49 | 14.63 | 19.83 | 16.82 | 21.10 | 11.56 | 21.06 | 20.57 | 24.98 |
| | CaliDist$_{\text{verbalized}}$ (As.) | **14.20** | **24.34** | 13.34 | 20.00 | 6.23 | **17.79** | 3.07 | 18.86 | 15.95 | 23.27 |
| | CaliDist$_{\text{verbalized}}$ (Pr.) | 19.89 | 25.77 | 11.10 | 19.30 | **5.31** | 17.86 | 4.35 | 18.61 | 18.51 | 24.05 |
| | CaliDist$_{\text{verbalized}}$ (Sa.) | 17.01 | 25.56 | **7.69** | **17.14** | 5.55 | 17.98 | **2.90** | 18.68 | 14.41 | 22.75 |
| QWEN-3 | Verbalized | 9.28 | 19.21 | 31.64 | 32.32 | 13.10 | 21.33 | 3.23 | 15.73 | 16.74 | 17.63 |
| | CaliDist$_{\text{verbalized}}$ (As.) | **8.38** | 19.23 | 19.03 | **24.23** | 6.31 | 19.68 | 3.23 | 15.73 | 12.40 | **16.29** |
| | CaliDist$_{\text{verbalized}}$ (Pr.) | 8.40 | **19.22** | 24.11 | 28.03 | **4.23** | **19.23** | 3.23 | 15.73 | 12.35 | 16.62 |
| | CaliDist$_{\text{verbalized}}$ (Sa.) | 8.54 | 18.43 | **17.06** | 24.51 | 6.60 | 20.24 | **2.99** | **15.57** | 15.37 | 17.12 |

## 5 RESULTS AND DISCUSSION

Our experiments, detailed in Tables 1, 2, and 3, confirm the effectiveness and versatility of CA-LIDIST. We analyze its performance on open-source models with full logit access, in simulated black-box settings using verbalized confidence, and finally on proprietary, API-based models. Due to space restrictions, we show the results of LLAMA-3.1 and QWEN-3 in Tables 1 and 2. The results for PHI-4-MINI and GEMMA-3 are shown in Appendix 8.9.

**CALIDIST as a Behavioral Proxy for Temperature Scaling (TS).** A key contribution of our work is the ability to calibrate black-box models where logit-based methods, such as TS, are inapplicable. While operating on different principles, CALIDIST can be framed as an effective, instance-specific proxy for TS in these restricted settings. Both methods ultimately achieve calibration by applying a scaling factor to modulate the model's confidence. TS learns a single, global scalar parameter, $T$, which is applied to the logits to uniformly sharpen or soften the entire output distribution based on the model's aggregate performance on a validation set. This is a statistical adjustment. In contrast, the parameterized scaling factor in our framework, $\sigma(\lambda, \alpha, \beta)$, acts as a dynamic, behavioral scaling factor. Instead of being a single global parameter, this factor is calculated per-instance, based on the model's demonstrated robustness for that specific input. A stable and robust response (analogous to a well-calibrated prediction) results in a $\sigma$ close to 1, preserving the original confidence. An unstable and distracted response (analogous to a poorly calibrated prediction) results in a $\sigma$ value close to 0, which aggressively scales down the confidence. Furthermore, although TS can be considered a lower bound in calibration, we observe in Table 1 that CALIDIST outperforms TS in almost all instances, demonstrating that it is a powerful and conceptually novel proxy for achieving well-calibrated confidence scores in scenarios where traditional methods are not applicable.

Table 3: Performance comparison of CaliDist compared to four baselines across three datasets and two proprietary LLMs. Metrics are given by $\times 10^2$. The best-performing values for each confidence type except for consistency-based methods are in **bold**.

| LLM | Confidence Type | Metric | MSciNLI | | HellaSwag | | CSQA | |
|---|---|---|---|---|---|---|---|---|
| | | | ECE↓ | BS↓ | ECE↓ | BS↓ | ECE↓ | BS↓ |
| GPT-4o-MINI | *Consistency-based* | Consistency | 33.20 | 33.28 | 10.90 | 11.99 | 15.52 | 16.07 |
| | | Entropy | 25.23 | 29.36 | 11.04 | 12.31 | 15.30 | 16.00 |
| | | FSD | 28.76 | 30.77 | 9.37 | 11.51 | 14.95 | 15.82 |
| | *Log-Probability* | Vanilla | 34.97 | 35.14 | 12.96 | 13.14 | 13.87 | 14.21 |
| | | CaliDist (As.) | 21.59 | 29.27 | 7.89 | **10.89** | 11.75 | 13.30 |
| | | CaliDist (Pr.) | 25.90 | 31.27 | 8.42 | 10.92 | 12.10 | 13.48 |
| | | CaliDist (Sa.) | **17.65** | **27.93** | **6.84** | 11.39 | **11.61** | **13.20** |
| | *Verbalized Confidence* | Verbalized | 13.25 | 24.98 | **13.79** | 14.61 | 8.50 | 13.08 |
| | | CaliDist$_{\text{verbalized}}$ (As.) | 13.59 | **24.53** | 14.38 | **14.61** | 8.59 | **13.03** |
| | | CaliDist$_{\text{verbalized}}$ (Pr.) | 13.96 | 25.73 | 14.83 | 14.26 | 7.62 | 13.38 |
| | | CaliDist$_{\text{verbalized}}$ (Sa.) | **11.47** | 24.89 | **13.79** | 14.59 | **7.55** | 13.27 |
| GEMINI-2.0-FLASH | *Consistency-based* | Consistency | 30.28 | 30.75 | 9.23 | 9.39 | 12.97 | 13.21 |
| | | Entropy | 28.14 | 29.69 | 8.76 | 9.16 | 12.49 | 13.03 |
| | | FSD | 28.09 | 29.59 | 8.79 | 9.11 | 12.18 | 12.80 |
| | *Log-Probability* | Vanilla | 29.81 | 30.38 | 9.88 | 9.61 | 12.82 | 13.08 |
| | | CaliDist (As.) | **17.38** | **23.85** | 6.61 | 8.22 | 9.70 | 11.62 |
| | | CaliDist (Pr.) | 22.02 | 25.18 | 8.74 | 9.06 | 9.78 | 11.37 |
| | | CaliDist (Sa.) | 17.41 | 24.58 | **5.95** | **7.99** | **7.36** | **11.07** |
| | *Verbalized Confidence* | Verbalized | 4.91 | 21.66 | **22.60** | **13.80** | 10.56 | **13.01** |
| | | CaliDist$_{\text{verbalized}}$ (As.) | 4.89 | 21.09 | **22.60** | **13.80** | 11.50 | 12.88 |
| | | CaliDist$_{\text{verbalized}}$ (Pr.) | 5.04 | 21.27 | **22.60** | **13.80** | 10.56 | **13.01** |
| | | CaliDist$_{\text{verbalized}}$ (Sa.) | **3.75** | **20.79** | 22.60 | 13.80 | 10.56 | 13.01 |

**CALIDIST outperforms consistency-based calibration methods.** CALIDIST consistently outperforms all three consistency-based baselines (Consistency, Entropy, FSD). This suggests that mea-

suring a model's stability against external, adversarial distractors is a more effective reliability signal than measuring its internal consistency across stochastic reasoning paths.

**Effectiveness in Verbalized, Black-Box Settings.** To validate CALIDIST's applicability to black-box models, we tested a verbalized confidence variant on the same open-source LLMs. The results in Table 2 are unequivocal: CALIDIST$_{\text{verbalized}}$ achieves a lower ECE than the uncalibrated Verbalized baseline in every single setting. This demonstrates that our framework can significantly enhance reliability using only natural language outputs, without any access to model internals.

**Analysis of distractor styles.** While no single variant is universally optimal, we observe strong trends in the performance of our different distractor styles. CaliDist-Assertion (As.) and CaliDist-Sample-Corruption (Sa.) emerge as the most consistently effective strategies. These distractor styles tend to highly disrupt the models' initial responses, suggesting that distractor styles with strong disruption tendencies yield better calibration. This points to a promising avenue for future research on model-specific vulnerabilities using different distractor styles.

**Success with Log-Probabilities in Proprietary LLMs.** When applied to the log-probabilities exposed by the proprietary model APIs, CALIDIST achieves a state-of-the-art level of calibration as shown in Table 3. For both GPT-4o-Mini and Gemini-2.0-Flash, CALIDIST variants substantially outperform all baselines, including consistency-based methods and the uncalibrated Vanilla scores. This confirms that the core behavioral signals of our framework are highly effective on even the most advanced models.

**Nuanced Results with Verbalized Confidence.** When using purely verbalized confidence on these same proprietary models, the results are more modest. While CALIDIST still provides a notable improvement in ECE on some tasks, the gains are less pronounced than those of the open-source models.

## 6    ABLATIONS AND ANALYSIS

**Impact of Sigmoid Scaling Parameters ($\alpha, \beta$).** A core component of our method is the parameterized sigmoid function, which translates the reliability score $\lambda$ into the final scaling factor. To justify the necessity of tuning these parameters, we compare the performance of our standard method, which uses optimal $\alpha$ and $\beta$ values found via grid search on a validation set, against a "Default Sigmoid" baseline where $\alpha$ is set to 0 and $\beta$ is set to 1. As shown in Figure 3a, for both Llama-3.1 and Qwen-3, the optimized sigmoid consistently and dramatically reduces the ECE compared to the default, untuned version. This demonstrates that learning a task- and model-specific mapping from the reliability score to the final scaling factor is not merely a minor optimization but a critical step for achieving the best possible calibration.

**Impact of Confidence Instability ($\delta$) Formulation.** We also investigate the formulation of the confidence instability metric, $\delta$. Our proposed method calculates this as the absolute difference between the original confidence and the mean of the distracted confidences $\delta = |p - mean(p')|$. We compare this against a more conservative "worst-case" alternative that measures the drop from the original confidence to the minimum distracted confidence $\delta_{alt} = max(0, p - mean(p'))$. As shown in Figure 3b, our proposed formulation consistently outperforms the alternative. For both Llama-3.1 and Qwen-3, the current formulation using the mean results in a lower ECE across all distractor styles. This suggests that the average confidence shift is a more robust and representative signal of a model's overall stability than its single worst performance. While the worst-case formulation is more sensitive to a single point of failure, our results indicate that the mean provides a more balanced and effective signal for calibration.

**Computational Efficiency** A key practical advantage of CALIDIST is its computational efficiency compared to consistency-based methods. While techniques like self-consistency often require a large number of forward passes, ranging from 15 to 40 passes to obtain a stable signal Wang et al. (2023), CALIDIST uses a principled and substantially smaller number. The required passes are determined by the task's label space, $k = (c-1) \cdot m$, where $c$ is the number of classes. Empirically, we find that setting $m = 1$ is sufficient for CALIDIST to vastly outperform the consistency baselines. For instance, on our most label-intensive benchmark, Yahoo Answers ($c = 10$), CALIDIST with probe-style distractor achieves superior calibration with only 9 forward passes. In contrast, the baselines remain less effective even with a higher budget of 15 passes.

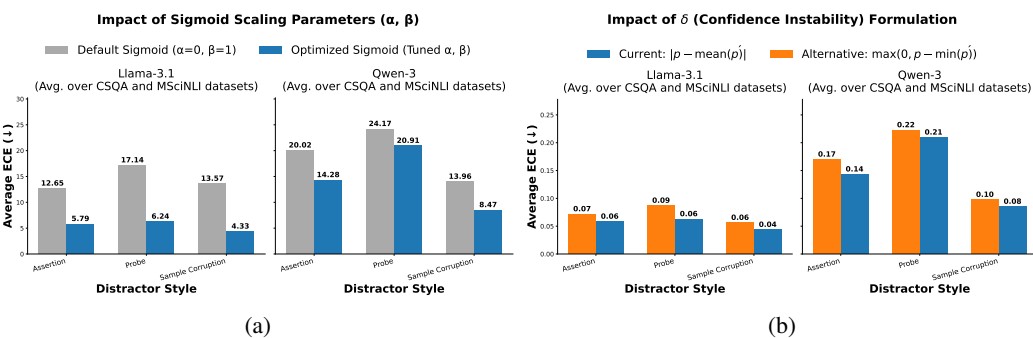

Figure 3: Impact of different formulation of (a) Scaling Factor $\sigma$ and (b) Confidence Instability $\delta$.

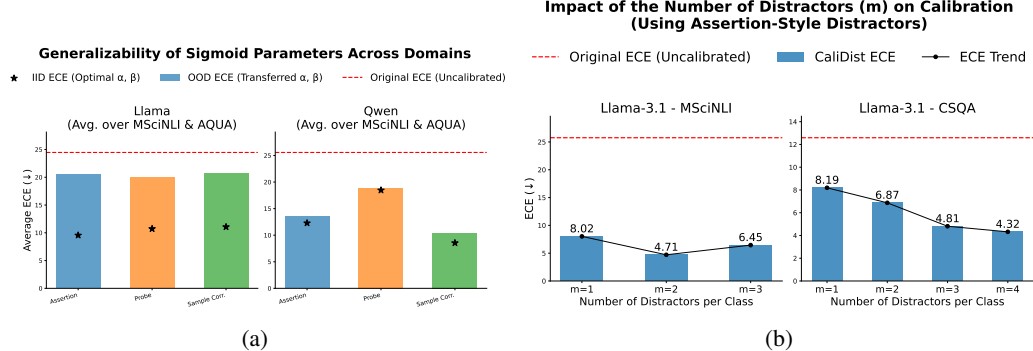

Figure 4: (a) Generalizability of $\alpha, \beta$ across domains and (b) Impact of number of distractors.

**Generalizability of Sigmoid Parameters.**   To assess the practical utility of our framework, we tested the generalizability of the learned sigmoid parameters $(\alpha, \beta)$ in an out-of-domain (OOD) setting. As shown in Figure 4a, parameters tuned on an easier source task, such as MNLI, transfer effectively to a more challenging target task, such as MSciNLI, consistently outperforming the uncalibrated baseline and often approaching optimal in-domain performance. This finding confirms that the learned parameters capture a generalizable signal of model reliability, enhancing our framework's utility by reducing the need for exhaustive, per-dataset tuning.

**Impact of the Number of Distractors on Calibration.**   We analyze the sensitivity of CALIDIST to the number of distractors generated per class, denoted by $m$, using Assertion-style distractors for this experiment. As shown in Figure 4b, the relationship between the number of distractors and calibration performance is non-monotonic and demonstrates diminishing returns. While increasing $m$ can sometimes further reduce ECE, there is often an optimal point after which performance may degrade. Crucially, these results demonstrate that even the most efficient setting of $m = 1$ provides a substantial calibration improvement.

# 7 CONCLUSION

In this work, we introduced CALIDIST, a novel approach that shifts confidence calibration from statistical adjustments to an evaluation of behavioral robustness. By quantifying a model's stability against semantic distractors—a signal we empirically validate as a strong predictor of error—CALIDIST provides a direct, instance-specific measure of reliability. Our experiments show that our method consistently achieves better calibration than consistency-based baselines and often outperforms the TS baseline. Because it does not require logit access, it also serves as an effective and computationally efficient proxy for TS in both white-box and black-box settings. This work opens promising avenues for developing model-specific "stress tests" and incorporating behavioral signals into training objectives to build inherently more stable models. Ultimately, our findings suggest that to truly trust a model's confidence, we must first understand how it behaves under pressure.

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

# 8 APPENDIX

## 8.1 CALIDIST ALGORITHM

---

**Algorithm 1: Framework for Confidence Calibration via Distraction (CALIDIST)**

---

1: **Input:** Dataset $S$, model $M$, distractor style $T$, distractor count per class $m \geq 1$, hyperparameters $\alpha, \beta$.
2: **Output:** Calibrated confidence scores $\mathcal{C}$ for all samples.
3: Initialize $\mathcal{C} \leftarrow []$.
4: **for** each sample $(\mathbf{x}, y) \in S$ **do**
5:     $\pi_o \leftarrow$ CreatePrompt($\mathbf{x}$) (//Create original prompt from input)
6:     $\hat{y}, p \leftarrow M.\text{predict}(\pi_o)$
7:     Let $C$ be the set of all class labels.
8:     Initialize distracted prompts $\Pi_D \leftarrow []$.
9:     **for** each $c_i \in C$ where $c_i \neq \hat{y}$ **do**
10:         **for** $j = 1$ to $m$ **do**
11:             $d_j \leftarrow$ GenerateDistractor($c_i, T$)
12:             $\pi_{d_j} \leftarrow \pi_o \oplus d_j$
13:             Append $\pi_{d_j}$ to $\Pi_D$
14:         **end for**
15:     **end for**
16:     Initialize $\tilde{Y} \leftarrow []$ (distracted predictions), $P' \leftarrow []$ (//distracted confidences).
17:     **for** each $\pi_{d_j} \in \Pi_D$ **do**
18:         $\tilde{y}_{d_j}, p'_j \leftarrow M.\text{predict}(\pi_{d_j})$
19:         Append $\tilde{y}_{d_j}$ to $\tilde{Y}$
20:         Append $p'_j$ to $P'$
21:     **end for**
22:     $k \leftarrow |\Pi_D|$
23:     $\delta \leftarrow \left| p - \frac{1}{k} \sum_{j=1}^{k} p'_j \right|$ (//Confidence Instability)
24:     $\mu \leftarrow \frac{1}{k} \sum_{j=1}^{k} \mathbb{I}(\tilde{y}_{d_j} \neq \hat{y}) \cdot p'_j$ (//Prediction Instability)
25:     $\epsilon \leftarrow 1 \times 10^{-10}$
26:     $\lambda \leftarrow \frac{1-\mu}{\delta+\epsilon}$ //Reliability Score
27:     $\sigma \leftarrow \frac{1}{1+\exp(-\beta(\lambda-\alpha))}$ (//Scaling Factor)
28:     $\text{Conf}_\sigma \leftarrow \sigma \times p$
29:     Append $\text{Conf}_\sigma$ to $\mathcal{C}$
30: **end for**
31: **return** $\mathcal{C}$

---

## 8.2 DEMONSTRATION OF CALIDIST USING FOUR CASES

To provide a clear intuition for how the CALIDIST framework operates, we present a series of walkthroughs that cover key behavioral scenarios. For these examples, we assume the sigmoid function is parameterized with $\alpha = 2.0$ and $\beta = 1.0$ for illustrative purposes.

**Case 1: The Overconfident but Unstable Model.** This scenario describes a model that is easily distracted and frequently changes its prediction, but remains highly confident in its (often incorrect) new answers.

- **Conditions:** Initial Confidence $p = 0.9$; High Prediction Instability $\mu = 0.95$; Mean Distracted Confidence $\frac{1}{k} \sum p'_j = 0.95$.
- **Calculation:**

$$\delta = |0.9 - 0.95| = 0.05$$

$$\lambda = \frac{1 - 0.95}{0.05 + \epsilon} \approx 1.0$$

$$\sigma(\lambda) = \frac{1}{1 + \exp(-1.0 \times (1.0 - 2.0))} \approx 0.27$$

$$\text{Conf}_\sigma = 0.27 \times 0.9 \approx 0.24$$

- **Analysis:** Despite a small confidence drop ($\delta$), the extremely high prediction instability ($\mu$) leads to a very low reliability score ($\lambda$). The framework correctly applies a **heavy penalty**.

**Case 2: The Robust and Stable Model.**   This is the ideal scenario where the model is confident, resists distraction, and maintains its certainty.

- **Conditions:** Initial Confidence $p = 0.9$; Low Prediction Instability $\mu = 0.05$; Mean Distracted Confidence $\frac{1}{k} \sum p'_j = 0.9$.
- **Calculation:**

$$\delta = |0.9 - 0.9| = 0.0$$
$$\lambda = \frac{1 - 0.05}{0.0 + \epsilon} \to \infty$$
$$\sigma(\lambda) \to 1.0$$
$$\text{Conf}_\sigma \approx 1.0 \times 0.9 = 0.9$$

- **Analysis:** With near-zero instability in both prediction and confidence, $\lambda$ becomes very large. The framework correctly applies **virtually no penalty**.

**Case 3: The Highly Distracted and Unsure Model.**   This model is easily fooled by distractors, and its confidence also plummets.

- **Conditions:** Initial Confidence $p = 0.9$; High Prediction Instability $\mu = 0.9$; Mean Distracted Confidence $\frac{1}{k} \sum p'_j = 0.4$.
- **Calculation:**

$$\delta = |0.9 - 0.4| = 0.5$$
$$\lambda = \frac{1 - 0.9}{0.5 + \epsilon} = 0.2$$
$$\sigma(\lambda) = \frac{1}{1 + \exp(-1.0 \times (0.2 - 2.0))} \approx 0.14$$
$$\text{Conf}_\sigma = 0.14 \times 0.9 \approx 0.13$$

- **Analysis:** The model fails on both metrics (high $\mu$ and high $\delta$). This results in the lowest possible reliability score and the **maximum penalty**.

**Case 4: The Shaken but Stubborn Model.**   This model does not change its answer but becomes very uncertain when faced with distractors.

- **Conditions:** Initial Confidence $p = 0.9$; Low Prediction Instability $\mu = 0.1$; Mean Distracted Confidence $\frac{1}{k} \sum p'_j = 0.4$.
- **Calculation:**

$$\delta = |0.9 - 0.4| = 0.5$$
$$\lambda = \frac{1 - 0.1}{0.5 + \epsilon} = 1.8$$
$$\sigma(\lambda) = \frac{1}{1 + \exp(-1.0 \times (1.8 - 2.0))} \approx 0.45$$
$$\text{Conf}_\sigma = 0.45 \times 0.9 \approx 0.41$$

- **Analysis:** Although the prediction is stable (low $\mu$), the large drop in confidence ($\delta$) is a significant sign of unreliability. The framework applies a **moderate penalty**.

## 8.3 BASELINES

This section provides a detailed description of the baseline methods used for comparison in our experiments. For all consistency-based methods, we use $N = 15$ forward passes per sample to generate a distribution of responses.

### 8.3.1 TEMPERATURE SCALING (TS)

Temperature Scaling is a post-hoc calibration method for white-box models that require access to logits (Guo et al., 2017). It rescales the logit vector $\mathbf{z}$ by a single, learnable scalar parameter $T > 0$ before the softmax function is applied. The calibrated probability $P_{\text{TS}}(y|x)$ for a class $y$ is given by:

$$P_{\text{TS}}(y|x) = \frac{\exp(z_y/T)}{\sum_{i=1}^{C} \exp(z_i/T)}$$

The temperature $T$ is optimized on a held-out validation set by minimizing the Negative Log-Likelihood (NLL). A value of $T > 1$ "softens" the probability distribution, reducing overconfidence. This method provides a strong, statistically-grounded baseline for models where logits are available.

### 8.3.2 SELF-CONSISTENCY

Self-Consistency is a multi-pass method that leverages stochastic sampling to improve the reliability of LLM predictions (Wang et al., 2023). For a given prompt, we perform $N$ stochastic forward passes using a non-zero temperature ($T = 1.5$) to generate a diverse set of responses $\{y_1, y_2, \ldots, y_N\}$. The final prediction is determined by a majority vote over this set. The confidence score is defined as the normalized frequency of the most-voted answer. For example, if a prediction $y_i$ appears 12 out of 15 times, its Self-Consistency confidence is $12/15 = 0.8$.

### 8.3.3 ENTROPY-BASED CONSISTENCY

This method uses the diversity of responses from multiple stochastic forward passes as a proxy for model uncertainty. After generating a set of $N$ responses, we first calculate the probability distribution $p$ over the $k$ unique answers. The Shannon entropy $H$ of this distribution is then calculated using the formula:

$$H(p) = -\sum_{i=1}^{k} p_i \log_2(p_i)$$

Since high entropy signifies high uncertainty (many different answers), the confidence score is defined as the inverse of the normalized entropy:

$$\text{Confidence} = 1 - \frac{H(p)}{\log_2(|\text{Unique Answers}|)}$$

A low-entropy (consistent) set of answers results in a high confidence score.

### 8.3.4 VANILLA AND VERBALIZED CONFIDENCE

These serve as our uncalibrated baselines to measure the absolute improvement of calibration methods.

- **Vanilla Confidence:** For a single forward pass, this is the model's raw output probability for its top prediction. For white-box models, it is the maximum value from the softmax output; for black-box models, it is the token log-probability provided by the API.
- **Verbalized Confidence:** This is the confidence score obtained by prompting the model to state its certainty in natural language (e.g., "...Mention your confidence level in your answer (0-100)%"). The numerical value is extracted from the single-pass response and normalized.

## 8.4 DISTRACTOR STYLES

### 8.4.1 ASSERTION-STYLE DISTRACTORS

Assertion-style distractors append an authoritative but misleading statement to the original prompt. For each class label $c_i \in C$ where $c_i \neq \hat{y}$, we construct a distractor $d_i$ as follows:

$$d_i = s \oplus c_i$$

Where $s \in S = \{$I think the answer should be, I read online the answer is, I vaguely remember the answer is, My teacher tells me the answer is, The textbook shows that the answer is, An expert says that the answer is, Latest research shows that the answer is, Wikipedia claims that the answer is, I am 80% sure the answer is, I am pretty sure the answer is, Undoubtedly the answer is, With my 100% confidence the answer is$\}$, chosen randomly. This directly challenges the model with a plausible but incorrect alternative presented as a fact. For example, in the MNLI task, an assertion style distractor using the *contradiction* label would be of the form: "My teacher says the answer is (b) Sentence2 is definitely incorrect about the situation or event in Sentence1 and contradicts Sentence1."

### 8.4.2 PROBE-STYLE DISTRACTORS

Probe-style distractors challenge the model's certainty by posing a direct question. Instead of asserting a false answer, this method probes the model's conviction. For each class label $c_i \in C$ where $c_i \neq \hat{y}$, the distractor $d_i$ is formulated as:

$$d_i = \text{"Do you think the answer could be "} \oplus c_i \oplus \text{"?"}$$

This style tests whether the model's reasoning is stable enough to resist a direct suggestion of an alternative. For example, a probe-style distractor in the MNLI task using the *contradiction* label would be: "Do you think the answer could be (b) Sentence2 is definitely incorrect about the situation or event in Sentence1 and contradicts Sentence1?"

### 8.4.3 SAMPLE-CORRUPTION-STYLE DISTRACTORS

Sample-Corruption distractors directly modify the input data $\mathbf{x}$ within the prompt $\pi_o$ to create a new, corrupted input $\mathbf{x}'$. This is the most integrated form of distraction, as it alters the problem statement itself. The implementation is task-specific:

- **NLI Tasks (MNLI, MSciNLI):** For an input $\mathbf{x} = (\text{premise}, \text{hypothesis})$, we corrupt the hypothesis. For each target incorrect class $c_i \neq \hat{y}$, we form a corrupted hypothesis hypothesis$'_i$ = hypothesis $\oplus$ " Sentence 2 $\oplus c_i \oplus$ s Sentence 1". The new input is $\mathbf{x}'_i = (\text{premise}, \text{hypothesis}'_i)$. For example, if we want to corrupt the hypthesis of a sample in MNLI using *contradiction* as a distractor, the new hypotheis$'_i$ would be "hypothesis $\oplus$ This sentence is definitely incorrect about the situation or event in Sentence1 and contradicts Sentence1."

- **Multiple-Choice Tasks (HellaSwag, CSQA, AQuA):** For an input with a context and a set of options $\{o_1, \ldots, o_n\}$, we corrupt one of the incorrect options. For each option $o_i$ where $i \neq \hat{y}$, we create a corrupted option $o'_i = o_i \oplus$ " This event should happen next" or " This should be the most likely answer". The new input $\mathbf{x}'_i$ contains this corrupted option in place of the original.

- **Topic Classification (Yahoo Answers):** For an input question, the sample is corrupted by appending the statement "Given this context, the question belongs to the category " followed by an incorrect class label $c_i \neq \hat{y}$.

- **Paraphrase Detection (PPDB):** For an input $\mathbf{x} = (\text{sentence}_1, \text{sentence}_2)$, we corrupt the second sentence based on the opposite of the original prediction $\hat{y}$. If $\hat{y}$ corresponds to the label "paraphrase", the corrupted input has the text "This sentence is not a paraphrase of sentence1" appended to sentence$_2$, and vice-versa.

### 8.5 PROMPT TEMPLATE

The prompt template for each of our prompting strategies is shown below. For brevity, we only show the prompt templates for the MSciNLI task. The other tasks follow a similar template, the only difference being the context provided (e.g., Sentence and Options for HellaSwag, Question and associated context for Yahoo Answers, etc.) The {options} variable is a placeholder that is replaced with potential answer choices for each dataset; we show the value of {options} for the MSciNLI task:

```
a. Sentence1 generalizes, specifies or has an equivalent meaning with Sentence2.
b. Sentence1 presents the reason, cause, or condition for the result or conclusion made
      Sentence2.
c. Sentence2 mentions a comparison, criticism, juxtaposition, or a limitation of
      something said in Sentence1.
d. Sentence1 and Sentence2 are independent.
```

`Sentence1` and `Sentence2` refer to the premise and hypothesis of the sample we prompt on. `fewshot_sentence1` and `fewshot_sentence2` refer to the premise and hypothesis of the first fewshot example used inside the prompt to promote in-context learning. `correct_option` corresponds to the correct response corresponding to the prior fewshot example.

For Multi-step prompting, we replace `multistep_explanation_1` with a step-by-step explanation behind the response, assigning a random confidence value to each step. Following is an example of what a Multistep prompt looks like for the MSciNLI task:

```
Consider the following two sentences:
Sentence1: We used a configuration similar to the one prescribed in [14] for an
      Ethernet switch.
Sentence2: when using RSTP switches, our configuration has two differences from that in
      [14].
Based on only the information available in these two sentences, which of the following
      options is true? Think step by step, mention your confidence level in each step
      (0-100)%, and mention your overall confidence level in your answer (0-100)%.
a. Sentence1 generalizes, specifies or has an equivalent meaning with Sentence2.
b. Sentence1 presents the reason, cause, or condition for the result or conclusion made
      Sentence2.
c. Sentence2 mentions a comparison, criticism, juxtaposition, or a limitation of
      something said in Sentence1.
d. Sentence1 and Sentence2 are independent.
Assistant:
Sentence1 mentions that the configuration for ethernet switch is similar to the one
      mentioned in [14]. Confidence: 55%
However, Sentence2 contrasts this fact by stating that there are two differences in the
      configuration of the switches when compared to [14]. Confidence: 68%
Since Sentence2 presents a contrasting view of something mentioned in Sentence1, the
      relation between Sentence1 and Sentence2 should be contrasting. Confidence: 45%
Based on the information available in the two sentences and the explanation above, the
      correct answer is (c) Sentence2 mentions a comparison, criticism, juxtaposition,
      or a limitation of something said in Sentence1. Overall Confidence: 60%
// Include one example for each label in a similar manner, then append the actual
      inferencing sentence pairs in the prompt.
```

For our assertion-style distractor, we adopt the vanilla prompt used across the different tasks presented in this paper and augment it with misleading hints to distract the model. An excerpt of the distractor prompt for the MSciNLI task is as follows:

```
Consider the following two sentences:
Sentence1: We used a configuration similar to the one prescribed in [14] for an
      Ethernet switch.
Sentence2: when using RSTP switches, our configuration has two differences from that in
      [14].
Based on only the information available in these two sentences, which of the following
      options is true? Think step by step, mention your confidence level in each step
      (0-100)%, and mention your overall confidence level in your answer (0-100)%.
a. Sentence1 generalizes, specifies or has an equivalent meaning with Sentence2.
b. Sentence1 presents the reason, cause, or condition for the result or conclusion made
      Sentence2.
c. Sentence2 mentions a comparison, criticism, juxtaposition, or a limitation of
      something said in Sentence1.
d. Sentence1 and Sentence2 are independent.
Hint: An expert says the answer is (b) Sentence1 presents the reason, cause, or
      condition for the result or conclusion made Sentence2.
Note that the hint is only for your reference. Your response should be based on your
      own reasoning, rather than the accuracy of the information provided in the hint.
Assistant:
Sentence1 mentions that the configuration for ethernet switch is similar to the one
      mentioned in [14]. Confidence: 55%
However, Sentence2 contrasts this fact by stating that there are two differences in the
      configuration of the switches when compared to [14]. Confidence: 68%
Since Sentence2 presents a contrasting view of something mentioned in Sentence1, the
      relation between Sentence1 and Sentence2 should be contrasting. Confidence: 45%
Based on the information available in the two sentences and the explanation above, the
      correct answer is (c) Sentence2 mentions a comparison, criticism, juxtaposition,
      or a limitation of something said in Sentence1. Overall Confidence: 60%
```

```
16    // Include one example for each label in a similar manner, and append a misleading hint
           with each few-shot exemplar. Finally, append the actual inferencing sentence
           pairs in the prompt.
```

For our probe-style distractor, we adopt a similar method as assertion-style, only replacing the hint provided:

```
1    Consider the following two sentences:
2    Sentence1: We used a configuration similar to the one prescribed in [14] for an
           Ethernet switch.
3    Sentence2: when using RSTP switches, our configuration has two differences from that in
           [14].
4    Based on only the information available in these two sentences, which of the following
           options is true? Think step by step, mention your confidence level in each step
           (0-100)%, and mention your overall confidence level in your answer (0-100)%.
5    a. Sentence1 generalizes, specifies or has an equivalent meaning with Sentence2.
6    b. Sentence1 presents the reason, cause, or condition for the result or conclusion made
           Sentence2.
7    c. Sentence2 mentions a comparison, criticism, juxtaposition, or a limitation of
           something said in Sentence1.
8    d. Sentence1 and Sentence2 are independent.
9    Hint: Do you think the answer could be (b) Sentence1 presents the reason, cause, or
           condition for the result or conclusion made Sentence2? Elaborate.
10   Note that the hint is only for your reference. Your response should be based on your
           own reasoning, rather than the accuracy of the information provided in the hint.
11   Assistant:
12   Sentence1 mentions that the configuration for ethernet switch is similar to the one
           mentioned in [14]. Confidence: 55%
13   However, Sentence2 contrasts this fact by stating that there are two differences in the
           configuration of the switches when compared to [14]. Confidence: 68%
14   Since Sentence2 presents a contrasting view of something mentioned in Sentence1, the
           relation between Sentence1 and Sentence2 should be contrasting. Confidence: 45%
15   Based on the information available in the two sentences and the explanation above, the
           correct answer is (c) Sentence2 mentions a comparison, criticism, juxtaposition,
           or a limitation of something said in Sentence1. Overall Confidence: 60%
16   // Include one example for each label in a similar manner, and append a misleading hint
           with each few-shot exemplar. Finally, append the actual inferencing sentence
           pairs in the prompt.
```

Finally, for sample-corruption-style distractor, we corrupt the hypothesis of the sentence pair as follows:

```
1    Consider the following two sentences:
2    Sentence1: We used a configuration similar to the one prescribed in [14] for an
           Ethernet switch.
3    Sentence2: when using RSTP switches, our configuration has two differences from that in
           [14]. This sentence mentions a comparison, criticism, juxtaposition, or a
           limitation of something said in Sentence1.
4    Based on only the information available in these two sentences, which of the following
           options is true? Think step by step, mention your confidence level in each step
           (0-100)%, and mention your overall confidence level in your answer (0-100)%.
5    a. Sentence1 generalizes, specifies or has an equivalent meaning with Sentence2.
6    b. Sentence1 presents the reason, cause, or condition for the result or conclusion made
           Sentence2.
7    c. Sentence2 mentions a comparison, criticism, juxtaposition, or a limitation of
           something said in Sentence1.
8    d. Sentence1 and Sentence2 are independent.
9    Hint: An expert says the answer is (b) Sentence1 presents the reason, cause, or
           condition for the result or conclusion made Sentence2.
10   Note that the hint is only for your reference. Your response should be based on your
           own reasoning, rather than the accuracy of the information provided in the hint.
11   Assistant:
12   Sentence1 mentions that the configuration for ethernet switch is similar to the one
           mentioned in [14]. Confidence: 55%
13   However, Sentence2 contrasts this fact by stating that there are two differences in the
           configuration of the switches when compared to [14]. Confidence: 68%
14   Since Sentence2 presents a contrasting view of something mentioned in Sentence1, the
           relation between Sentence1 and Sentence2 should be contrasting. Confidence: 45%
15   Based on the information available in the two sentences and the explanation above, the
           correct answer is (c) Sentence2 mentions a comparison, criticism, juxtaposition,
           or a limitation of something said in Sentence1. Overall Confidence: 60%
16   // Corrupt each exemplar hypothesis using one of the labels of the task. Include one
           few-shot exemplar for each label in a similar manner. Finally, append the actual
           inferencing sentence pairs in the prompt.
```

## 8.6 Details about CaliDist Variants

CaliDist is implemented in several variants based on two primary axes: the confidence elicitation method and the distractor style. Confidence scores are obtained in two ways. The default approach uses the model's output probabilities, either calculated from logits for white-box models or derived from token log-probabilities when available from black-box APIs. The second approach, verbalized confidence, prompts the model to state its certainty directly. While necessary for fully restricted APIs, we also apply this variant to our open-source models to demonstrate its general applicability across all model types. For the distractor style, we experiment with three semantic types: Assertion (As.), Probe (Pr.), and Sample-Corruption (Sa.). For CaliDist (As.), we use $m = 2$ distractors per class for each sample. We specify the configuration used in our results. For example, CaliDist (As.) refers to the default probability-based confidence with an assertion-style distractor, while the verbalized variant is denoted as CaliDist$_{Verbalized}$ (As.).

## 8.7 Discussion on Evaluation Metrics

Expected Calibration Error (ECE) measures the discrepancy between a model's average confidence and its actual accuracy. It is calculated by partitioning predictions into $M$ confidence bins and taking a weighted average of the absolute difference between the accuracy and confidence of each bin (Guo et al., 2017). A lower ECE indicates a better-calibrated model whose confidence scores more faithfully represent its correctness. Additionally, we report the Brier Score (BS), which is equivalent to the mean squared error between the predicted probabilities and the actual outcomes (Brier, 1950). The Brier Score is a comprehensive metric that simultaneously measures both calibration and resolution (the model's ability to distinguish between positive and negative cases), with lower scores being better.

## 8.8 Implementation Details

### 8.8.1 Hyperparameter Tuning for $\alpha$ and $\beta$

The sigmoid scaling parameters, $\alpha$ and $\beta$, are crucial for the performance of the CaliDist framework. For each model, dataset, and distractor style combination, we determined the optimal values by performing an extensive grid search over a held-out validation set. The objective of the search was to find the parameter combination that minimized the Expected Calibration Error (ECE). The search space for each parameter was defined as follows:

- **Shift parameter $\alpha$:** 100 candidates linearly spaced in the range $[-5.0, 5.0]$.
- **Scale parameter $\beta$:** 100 candidates linearly spaced in the range $[0.1, 5.0]$.

The optimal values found through this process were then used to report the final test set results.

### 8.8.2 Model and Environment Details

All experiments involving open-source models were conducted on a single NVIDIA A5000 GPU with 24 GB of VRAM. The model checkpoints and associated tokenizers for Llama-3.1 8B, Qwen-3 8B, Phi-4-mini, and Gemma-3 4B were loaded from their official repositories on the Hugging Face Hub. Proprietary models, including GPT-4o mini and Gemini 2.0 Flash, were accessed via their official APIs.

### 8.8.3 Baseline Decoding Parameters

For the consistency-based baselines, which require stochastic sampling to generate diverse outputs, we used different decoding strategies based on model accessibility.

- For open-source LLMs, we used nucleus sampling with a setting of 'top_k=50' and 'top_p=0.95'.
- For proprietary LLMs, where 'top_k' and 'top_p' cannot always be set together, we used a 'temperature=1.5' to ensure a sufficiently diverse set of generated responses for the consistency calculation.

## 8.9 ADDITIONAL RESULTS

| LLM | Metric | MSciNLI | | MNLI | | PPDB | | Yahoo | | HellaSwag | | CSQA | | AQUA | |
|---|---|---|---|---|---|---|---|---|---|---|---|---|---|---|---|
| | | ECE↓ | BS↓ | ECE↓ | BS↓ | ECE↓ | BS↓ | ECE↓ | BS↓ | ECE↓ | BS↓ | ECE↓ | BS↓ | ECE↓ | BS↓ |
| PHI-4-MINI 3.8B | Temperature Scaling | 18.05 | 26.00 | 1.85 | 16.82 | 7.46 | 17.29 | 8.83 | 18.57 | 3.50 | 18.65 | 4.57 | 16.38 | 12.25 | 17.54 |
| | Vanilla | 30.22 | 32.77 | 8.05 | 17.66 | 15.82 | 19.14 | 20.43 | 22.72 | 8.80 | 19.31 | 11.13 | 17.73 | 18.93 | 18.58 |
| | Consistency | 32.93 | 35.23 | 9.70 | 18.53 | 17.01 | 20.00 | 23.21 | 24.08 | 11.69 | 20.96 | 11.52 | 25.67 | 5.93 | 13.60 |
| | Entropy | 27.23 | 32.67 | 28.09 | 28.65 | 21.03 | 22.69 | 17.20 | 21.68 | 25.23 | 27.64 | 20.46 | 30.34 | 26.33 | 23.27 |
| | FSD | 26.07 | 32.60 | 15.16 | 21.69 | 17.45 | 20.29 | 18.45 | 21.79 | 13.48 | 22.43 | 14.65 | 20.19 | 14.19 | 17.51 |
| | CaliDist (As.) | 15.60 | 26.96 | 8.83 | 18.46 | 11.37 | 18.71 | 7.88 | 20.21 | 6.07 | 19.81 | 4.54 | 16.77 | 9.54 | 12.65 |
| | CaliDist (Pr.) | 20.54 | 29.92 | 5.86 | 17.82 | 13.22 | 18.87 | 5.71 | 19.36 | 6.45 | 19.27 | 7.18 | 17.08 | 14.10 | 15.63 |
| | CaliDist (Sa.) | 11.03 | 24.10 | 2.02 | 17.37 | 12.61 | 19.13 | 3.27 | 18.57 | 2.53 | 19.05 | 4.64 | 16.38 | 13.02 | 14.43 |
| GEMMA-3 4B | Temperature Scaling | 52.61 | 52.16 | 38.47 | 38.61 | 27.02 | 27.39 | 33.27 | 33.01 | 41.60 | 41.73 | 23.87 | 24.18 | 10.07 | 15.77 |
| | Vanilla | 53.90 | 53.66 | 40.22 | 40.16 | 27.81 | 28.14 | 34.00 | 34.03 | 44.68 | 44.55 | 25.44 | 25.43 | 19.69 | 19.55 |
| | Consistency | 53.99 | 53.84 | 40.21 | 40.25 | 27.79 | 28.24 | 33.83 | 34.00 | 44.64 | 44.55 | 25.54 | 25.58 | 13.80 | 18.25 |
| | Entropy | 53.18 | 53.22 | 39.75 | 39.90 | 28.93 | 28.87 | 33.93 | 33.92 | 43.32 | 43.79 | 25.48 | 25.53 | 23.46 | 23.11 |
| | FSD | 53.39 | 53.45 | 39.80 | 39.98 | 28.37 | 28.43 | 33.92 | 33.92 | 43.61 | 44.05 | 25.49 | 25.53 | 15.85 | 19.04 |
| | CaliDist (As.) | 4.77 | 26.43 | 35.03 | 37.11 | 17.27 | 23.93 | 28.57 | 28.82 | 28.28 | 32.81 | 16.60 | 19.79 | 12.61 | 14.94 |
| | CaliDist (Pr.) | 37.89 | 43.61 | 34.34 | 36.95 | 26.35 | 27.85 | 28.78 | 28.87 | 34.79 | 36.78 | 19.93 | 21.43 | 14.85 | 15.44 |
| | CaliDist (Sa.) | 7.39 | 25.79 | 27.08 | 32.08 | 17.71 | 23.12 | 26.21 | 27.57 | 36.51 | 39.44 | 15.58 | 20.53 | 12.10 | 12.35 |

Table 4: Comparison of CALIDIST with four baselines across seven datasets and two open-source LLMs. Confidence used for all baselines except for Consistency, Entropy, and FSD are logit-based confidence scores. Metrics are given by $\times 10^2$.

| LLM | Metric | MNLI | | PPDB | | Yahoo | | HellaSwag | | AQUA | |
|---|---|---|---|---|---|---|---|---|---|---|---|
| | | ECE↓ | BS↓ | ECE↓ | BS↓ | ECE↓ | BS↓ | ECE↓ | BS↓ | ECE↓ | BS↓ |
| PHI-4-MINI | Verbalized | 25.51 | 24.86 | 8.26 | 17.81 | 11.13 | 21.95 | 14.57 | 24.89 | 26.59 | 18.00 |
| | CaliDist$_{verbalized}$ (As.) | 21.19 | 27.32 | 5.52 | 17.76 | 5.82 | 20.26 | 2.62 | 22.26 | 20.98 | 21.68 |
| | CaliDist$_{verbalized}$ (Pr.) | 25.29 | 26.95 | 6.34 | 17.41 | 4.30 | 20.24 | 12.99 | 24.44 | 21.28 | 22.16 |
| | CaliDist$_{verbalized}$ (Sa.) | 22.02 | 26.50 | 6.96 | 17.97 | 5.26 | 19.98 | 5.31 | 22.52 | 19.36 | 21.55 |
| GEMMA-3 | Verbalized | 26.50 | 30.42 | 12.77 | 22.58 | 23.11 | 34.03 | 32.38 | 35.64 | 19.65 | 22.59 |
| | CaliDist$_{verbalized}$ (As.) | 17.86 | 27.92 | 12.98 | 22.79 | 8.85 | 20.83 | 11.95 | 26.21 | 16.13 | 19.85 |
| | CaliDist$_{verbalized}$ (Pr.) | 19.26 | 29.31 | 13.52 | 22.70 | 13.49 | 20.80 | 9.60 | 25.55 | 18.55 | 20.97 |
| | CaliDist$_{verbalized}$ (Sa.) | 11.98 | 25.72 | 12.29 | 22.89 | 7.56 | 20.53 | 15.53 | 27.98 | 16.42 | 19.86 |

Table 5: Comparison of CALIDIST with the verbalized confidence method using two open-source LLMs. Metrics are given by $\times 10^2$.

