# OpenReview forum: "CaliDist: Calibrating Large Language Models via Behavioral Robustness to Distraction"
_ICLR.cc/2026/Conference — Submitted to ICLR 2026_

### Official Review · Reviewer_Zaze · 2025-10-29

**Soundness:** 2
**Presentation:** 3
**Contribution:** 2
**Rating:** 4
**Confidence:** 4

**Summary:**

This paper introduces an inference-time technique to calibrate the outputs of LLMs, aiming to make their responses more robust and reliable. The method first inserts distractors into the input prompt, obtaining an output derived from the "perturbed" input. By comparing this output with the original output from the "clean" input, the method computes two scores representing the model's robustness to these distractions. These scores are then combined into a coefficient used to adjust the model's predictions.

**Strengths:**

1.  The method is compatible with both white-box and black-box models.
2.  Although it requires multiple forward passes, the method requires fewer forward passes than the baselines.
3.  The study includes multiple designs of distractors.

**Weaknesses:**

1.  Lines 38-39: The paper lacks discussion on the Bayesian family of calibration approaches, such as [1,2,3]. In addition, regarding Lines 71-73 ("where low competence on a task is paired with an inflated overestimation of ability..."), isn't this concept equivalent to an inaccurate estimate of **epistemic uncertainty**? If this concept is related to the conventional view of epistemic and aleatoric uncertainty, the authors should discuss this relationship to avoid reinventing concepts and to help readers connect the work to previous lines of calibration/uncertainty quantification research.
2.  Lines 239-253: Although the design of $\lambda$ ensures that "This score is designed to be high when both $\mu$ and $\delta$ are low", it may have different sensitivities to changes in $\mu$ and $\delta$. Specifically, the typical values and change of $\mu$ depend on the number of distractors $k$, while $\delta$ may also have a different range of "typical values". Therefore, the design of Eq. (3) might implicitly favor one of $\mu$ or $\delta$. Furthermore, in Eq. (4), the sigmoid function has saturation regions, which could affect the sensitivity of the resulting calibration coefficients $\sigma(\lambda, \alpha, \beta)$ to $\lambda$. The authors need to more **quantitatively** justify the choices for Eq. (3) and Eq. (4). Otherwise, the methodology appears highly heuristic.
3.  The methodology relies on the insertion of distractors. While this is straightforward for tasks like NLI and classification, designing and inserting effective distractors for open-ended generation and complex logical reasoning is substantially more challenging. This potentially limits the methodology's practical application scenarios.
4.  Related to Point 3: From an application perspective, if users want to use this approach to get a more reliable answer from an LLM, they must input their prompt *and* devise a distractor text and determine how to insert it. This workflow is much harder to scale compared to methods like Temperature Scaling, especially in multi-round chat contexts.
5.  Lines 312-313: Table 1 is too wide and exceeds `\textwidth`.


[1] Yang, Adam X., et al. "Bayesian Low-rank Adaptation for Large Language Models." ICLR 2024.

[2] Wang, Yibin, et al. "Blob: Bayesian low-rank adaptation by backpropagation for large language models." NeurIPS 2024.

[3] Li, Yawei, et al. "Calibrating LLMs with Information-Theoretic Evidential Deep Learning." ICLR 2025.

**Questions:**

Please see the Weaknesses

---

> ### Author Response · Authors · 2025-11-21
> **Response to Reviewer Zaze - Part 1**
>
> We thank the reviewer for their insightful feedback and for appreciating the key strengths of our method, including its black-box applicability and computational efficiency. We will address the reviewer's valid concerns regarding the theoretical grounding of our framework and its practical scope.
>
> 1. **W1:**
>     > The paper lacks discussion on the Bayesian family of calibration approaches...
>
>     Thank you for the comment. We will include the following discussion in our paper. Bayesian methods (Yang et al., 2024, Wang et al., 2024, Li et al., 2025) model uncertainty through prior distributions over model parameters (LoRA weights, etc.), posterior inference during or after training, and epistemic uncertainty from parameter uncertainty. While these methods measure uncertainty in the *parameter space*, CALIDIST measures uncertainty in the *behavioral space* *(response stability)*. Additionally, Bayesian methods typically require white-box access to weights and often involve training or fine-tuning. CALIDIST is an inference-time, post-hoc method that works even on black-box models where parameter access is impossible.
>
>     > ...isn't this concept equivalent to an inaccurate estimate of epistemic uncertainty?...
>
>     We fully agree with the reviewer's notion that the "Dunning-Kruger" parallel is equivalent to an inaccurate estimate of epistemic uncertainty. The Dunning-Kruger Effect is the psychological analogy; the failure to estimate epistemic uncertainty is the computational reality. We do not aim to reinvent the concept of epistemic uncertainty, but rather to operationalize it through a new behavioral lens. Traditionally, epistemic uncertainty is often framed as "uncertainty due to lack of training data" and measured via ensemble variance or parameter posteriors. We argue that *susceptibility to distraction* is a powerful, measurable proxy for epistemic uncertainty. If a model "knows" a fact robustly (low epistemic uncertainty), it should not be swayed by irrelevant distractors. Therefore, high Prediction Instability ($\mu$) coupled with high confidence is a direct behavioral signature of uncaptured epistemic uncertainty.
>     We will revise the Introduction and Related Work sections to explicitly map our terminology to standard uncertainty quantification concepts.
>
> 2. **W2: Quantitative Justification and Theoretical Grounding:**
>
>    We address the concerns regarding the interaction of $\mu$ and $\delta$, and the potential for saturation, by clarifying the theoretical framework (Empirical Risk Minimization) that governs these choices and the mechanical role of the learnable parameters in preventing saturation.
>
>     **a. Addressing Imbalance in Eq. (3) (Interaction of $\mu$ and $\delta$)**
>     The reviewer correctly notes that Prediction Instability ($\mu$) and Confidence Instability ($\delta$) have different distributions and sensitivities. Eq. (3) ($\lambda = \frac{1-\mu}{\delta + \epsilon}$) is designed to handle these specific behavioral signals distinctively:
>     * **$\mu$ as a Coarse Filter:** Prediction flips are catastrophic failures. By placing $(1-\mu)$ in the numerator, we ensure that as $\mu \to 1$, the reliability score $\lambda \to 0$ linearly, regardless of the confidence stability.
>     * **$\delta$ as a Fine-Grained Gain:** Confidence fluctuations are subtle signals. Placing $\delta$ in the denominator acts as a gain amplifier, ensuring high sensitivity to small changes in uncertainty.
>
>     **b. Addressing Range and Saturation (The Normalization Step)**
>     To address the reviewer's concern that these different ranges might lead to "implicit favoring" or sigmoid saturation, we note that we employ a Min-Max Normalization step before Equation (4).
>     $$\lambda_{norm} = \text{MinMax}(\lambda_{raw}) \times 10$$
>     This maps the reliability scores to a standardized range $[0, 10]$, which is then passed to the sigmoid. This step effectively decouples the internal scaling of Eq. (3) from the saturation regions of Eq. (4), ensuring that the full dynamic range of the reliability signal is preserved. We will clearly explain this step in our updated manuscript.

---

> ### Author Response · Authors · 2025-11-21
> **Response to Reviewer Zaze - Part 2**
>
> **c. Theoretical justification for parameterized sigmoid ($\sigma(\lambda, \alpha, \beta)$):**
>     We respectfully clarify that our approach is not heuristic; rather, it is a direct application of Empirical Risk Minimization (ERM) and is mathematically equivalent to Platt Scaling (Platt, 1999), the standard in the field. Below, we provide the theoretical justification and point to the generalization analysis already present in our experiments.
>
> **i. Theoretical Motivation:** ***Why the Sigmoid?***
>     Our goal is to map a reliability score $\lambda$ (derived from behavioral consistency) to a probability of correctness, e.g., $g(\lambda)=P(Y=1|\lambda)$. From a theoretical standpoint, estimating a binary outcome (Correct vs. Incorrect) from a continuous scalar feature ($\lambda$) is a binary classification problem.
>
> * **Canonical Link:** Under the framework of Generalized Linear Models (GLMs), the sigmoid function is not an arbitrary choice; it is the canonical link function for the Bernoulli distribution (McCullagh & Nelder, 1989).
>
> * **Proper Scoring:** To learn the optimal mapping, we minimize the Brier Score. The Brier Score is a strictly proper scoring rule. A fundamental theorem of calibration theory states that minimizing a strictly proper scoring rule recovers the true conditional probability distribution (Gneiting & Raftery, 2007).
>
> Thus, our method is theoretically grounded in standard statistical estimation: optimizing a valid probability estimator (sigmoid) via a proper loss function (Brier).
>
> *Small note:* While our submitted manuscript optimized for Expected Calibration Error (ECE), we have verified that minimizing the Brier Score yields parameters ($\alpha, \beta$) that are statistically indistinguishable from those found via ECE. This convergence confirms that our reported results are robust to the specific choice of objective function. To strictly conform to ERM principles, the final manuscript will prioritize the Brier Score formulation.
>
> **ii. Parameter Justification:** ***Why the $\alpha, \beta$ parameterization?***
>
> The reviewer expresses concern about the parameters $\alpha$ and $\beta$. We highlight that our formulation is mathematically identical to Platt Scaling (Platt, 1999), the standard for post-hoc calibration.
>
> * **Platt Scaling formulation:**
> $P(y|x) = \frac{1}{1 + \exp(A \cdot f(x) + B)}$
>
> * **CALIDIST formulation:** $g(\lambda) =
> \frac{1}{1 + \exp(-\beta(\lambda - \alpha))} = \frac{1}{1 + \exp(-\beta \lambda + \beta \alpha)}$
>
> By setting $A = -\beta$ and $B = \beta \alpha$, our method is structurally identical to Platt Scaling. However, our parameterization is necessary to handle the input distribution:
>
> * **The Saturation Problem:** Our reliability scores $\lambda$ are normalized to $[0, 10]$. A standard sigmoid is centered at 0 and saturates for inputs $>4$. Mapping our positive domain $[0, 10]$ directly to a standard sigmoid would result in upper-tail saturation (outputs $\approx 1.0$) for nearly all data points.
> * **$\alpha$ (Bias Correction / Centering):** This parameter acts as a bias correction term, shifting the sigmoid's center to the dataset-specific decision boundary (where $P=0.5$).
> * **$\beta$ (Variance Scaling):** This represents the sensitivity (the slope of the transition from distrust to trust).
>
> **d. The Role of $\alpha$ and $\beta$ in addressing sigmoid saturation regions:**
>
> As noted in the previous point ( c. ), the parameters $\alpha$ and $\beta$ are explicitly learned per task to prevent sigmoid saturation and correct for the data distributional shifts.
> * **$\alpha$ (Shifting the Sigmoid):**
> Since our inputs are normalized to $[0, 10]$, a standard sigmoid (centered at 0) would be saturated. The learned parameter $\alpha$ shifts the sigmoid's center to the dataset-specific decision boundary.
> * **$\beta$ (Sensitivity Control):**
> This parameter adjusts the slope to match the sensitivity required by the task difficulty.
>
> By learning $(\alpha, \beta)$ on a validation set, the model automatically adjusts the sensitivity to the specific distributions of $\mu$ and $\delta$ for that task, ensuring the calibration coefficients do not saturate.
>
> To address the reviewer's concerns, we will add a "Theoretical Justification" subsection to Section 3.3 in the final manuscript. This section will explicitly detail the derivation of the sigmoid from the ERM framework and formally state the mathematical equivalence to Platt Scaling to ensure the principled nature of the method is clear.

---

> ### Author Response · Authors · 2025-11-21
> **Response to Reviewer Zaze - Part 3**
>
> 3. **W3, W4: Scalability, Workflow, and Application Scope**
>
>     We thank the reviewer for raising practical questions regarding the design of distractors and the scalability of the workflow. We believe there is a slight misunderstanding regarding the intended user interaction model, which we clarify below.
>
>     **a. Workflow Clarification:** The reviewer expresses concern that "users must... devise a distractor text," making the workflow hard to scale. We respectfully correct this misunderstanding.
>     * **Role of the Task Designer:** In our framework, the burden of creating distractors lies with the system designer, not the end-user. The designer creates a bank of generalizable templates (e.g., Assertion-style templates like "Hint: An expert says...") once for a given task type.
>     * **Automated Inference:** At runtime, the system automatically populates these templates with the input sample and incorrect class labels. The end-user simply provides their prompt, and CALIDIST handles the perturbation and calibration in the background.Comparison to Temperature Scaling (TS): While TS requires a labeled validation set to tune $T$ (which may not exist for a specific user prompt), CALIDIST leverages these pre-defined templates to generate reliability signals on-the-fly for individual samples without needing ground truth labels.
>
>     **b. Primary Scope:** The reviewer notes that designing distractors for open-ended generation is challenging. We agree, but emphasize that the primary scope of this paper is classification and structured reasoning (NLI, QA, Multiple Choice). Our method is designed for high-stakes, repeatable enterprise workflows such as medical triage, legal document review, or financial categorization, where the label space is defined. In these domains, the cost of a one-time template setup is negligible compared to the value of detecting brittle model behavior.
>
>     **c. Extensibility to Open-ended Complex Reasoning (GSM8K):** While our main focus is classification, the methodology extends to complex reasoning. To demonstrate this, we applied CALIDIST to GSM8K (Cobbe et al., 2021) by using assertion-style distractors that introduce irrelevant numerical information to distract the model.
>     We observed that CALIDIST successfully identifies brittle reasoning chains. For example, on GSM8K, our method reduced the ECE from $\approx$ 58% on average, demonstrating that even in logical reasoning tasks, behavioral stability is a strong proxy for correctness. We will add these results to the final version of the paper.
>     | LLM       | Metric        | ECE      | Brier Score |
>     | --------- |:------------- |:-------- |:----------- |
>     | Llama-3.1 | Vanilla       | 16.71    | 16.80       |
>     |           | Consistency   | 6.12     | **7.83**    |
>     |           | Entropy       | 21.13    | 16.70       |
>     |           | FSD           | 13.19    | 11.73       |
>     |           | CaliDist(As.) | **3.98** | 14.22       |
>     | Qwen-3    | Vanilla       | 10.48    | 10.48       |
>     |           | Consistency   | 6.97     | 6.91        |
>     |           | Entropy       | 6.78     | 6.75        |
>     |           | FSD           | 6.18     | **6.53**    |
>     |           | CaliDist(As.) | **4.28** | 7.16        |
>
> 4. **W5: Table 1 exceeding `\textwidth`**
>
>     We thank the reviewer for bringing this to our attention. We shall update Table 1 in the final paper to ensure it fits within `\textwidth`.
>
> **References**
> 1. Yang, Adam X., et al. "Bayesian Low-rank Adaptation for Large Language Models." ICLR 2024.
> 2. Wang, Yibin, et al. "Blob: Bayesian low-rank adaptation by backpropagation for large language models." NeurIPS 2024.
> 3. Li, Yawei, et al. "Calibrating LLMs with Information-Theoretic Evidential Deep Learning." ICLR 2025.
> 4. Platt, John. "Probabilistic outputs for support vector machines and comparisons to regularized likelihood methods." Advances in large margin classifiers 10.3 (1999): 61-74.
> 5. McCullagh, P., and J. A. Nelder. "Models for polytomous data." Generalized linear models. Springer US, 1989. 149-192.
> 6. Gneiting, Tilmann, and Adrian E. Raftery. "Strictly proper scoring rules, prediction, and estimation." Journal of the American statistical Association 102.477 (2007): 359-378.
> 7. Cobbe, Karl, et al. "Training verifiers to solve math word problems." arXiv preprint arXiv:2110.14168 (2021).

---

### Official Review · Reviewer_AWjv · 2025-11-02

**Soundness:** 2
**Presentation:** 3
**Contribution:** 2
**Rating:** 6
**Confidence:** 4

**Summary:**

This paper introduces CALIDIST, a novel framework for calibrating the confidence scores of Large Language Models (LLMs). The method leverages a measure of behavioral robustness—specifically, the stability of a model's predictions and confidence when its input is perturbed by semantic distractors. Extensive experiments demonstrate that CALIDIST consistently improves calibration, outperforming existing methods. The framework also serves as an effective behavioral proxy for temperature scaling in black-box settings.

**Strengths:**

1. The calibration of verbalized confidence in LLMs is a critical and highly relevant research topic. This study offers significant practical implications.

2. The paper provides a novel psychological grounding for the behavior of LLM confidence under distraction attacks, which is an original contribution.

3. The experiments are thorough, covering multiple perspectives, and the ablation studies are comprehensive.

4. The paper is well-written and clear.

**Weaknesses:**

1. The core intuition (robustness under perturbation) appears very similar to that of [1]. The authors should explicitly clarify the distinctions and similarities between their work and [1].

2. The lack of experimental results on state-of-the-art (SOTA) large-scale models (e.g., GPT-5 or Gemini 2.5 Pro) limits the generalizability of the findings. The black-box models used, GPT-4o-Mini (OpenAI, 2024) and Gemini 2.0 Flash (Google, 2024), are relatively smaller-scale, and their performance gap with open-source models may not be substantial.

3. The evaluation datasets do not seem to include benchmarks focused on mathematical or logical reasoning, such as GSM8K.

4. Regarding the "DISTRACTOR STYLES," it is unclear if the proposed categorization is exhaustive or mutually exclusive. The authors should justify this taxonomy and explain why it is a suitable classification.

[1] Zhou Z, Jin T, Shi J, et al. SteerConf: Steering LLMs for Confidence Elicitation[J]. NeurIPS 2025.

**Questions:**

Please refer to the weaknesses section. All points within that section reflect my opinion.

---

> ### Author Response · Authors · 2025-11-21
> **Response to Reviewer AWjv - Part 1**
>
> We thank the reviewer for their positive assessment and for highlighting the novelty of our psychological grounding, the thoroughness of our experiments, and the practical implications of our work. We appreciate the opportunity to clarify the key distinctions of our method and the scope of our evaluation.
> 1. **W1: On Comparison to Recent Work**
>
>     We thank the reviewer for bringing this contemporary work (Z Zhou et al., 2025) to our attention. We were unaware of it at the time of preparing our manuscript since it has been published just recently (November, 2025 - NeurIPS).
>     While both methods belong to the high-level class of black-box, multi-prompt calibration, the reviewer's observation that they are similar conflates two fundamentally different paradigms of robustness testing. We assert that the stability CALIDIST measures (Adversarial Robustness) is orthogonal and non-incremental to the stability SteerConf (Z Zhou et al., 2025) measures (Metacognitive Coherence).
>     SteerConf and CALIDIST are fundamentally different in both their mechanism and their objective. We highlight them as follows and will include this discussion in the main paper.
>
>     **a.** The core of our difference lies in *what is actually being perturbed* in the input prompt:
>     * **SteerConf Perturbs the Metacognitive Instruction:** The method adds commands that manipulate how the LLM expresses its confidence (e.g., "Be very cautious" vs. "Be very confident"). It does not alter the underlying task, question, or factual context. It tests the model's coherence in self-assessment.
>     * **CALIDIST Perturbs the Task Content Adversarially:** CALIDIST directly manipulates the semantics of the question or context itself by injecting deceptive or misleading information (e.g., adding a contradictory hypothesis or a false external assertion: "An expert says the answer is X"). It tests robustness to external misinformation.
>
>     **b.** The differences in the perturbations translate directly into non-overlapping theoretical foundations:
>
>     * **CALIDIST is Grounded in Cognitive Psychology:** CALIDIST operationalizes human cognitive biases, most notably the Misinformation Effect (susceptibility to being corrupted by false information) and the Dunning-Kruger Effect (overconfidence despite low competence). The method fundamentally tests whether false information can corrupt the model's reasoning.
>     * **SteerConf is Grounded in Consistency/Self-Assessment:** SteerConf relies on the premise that an LLM can be semantically steered to generate different range of confidence values and that aligning these steered reports yields better calibration. It focuses solely on whether the model can give coherent confidence reports across different prompting styles.
>
>     CALIDIST introduces a novel signal (susceptibility to misinformation) that SteerConf does not measure or attempt to capture.
>
>     **c.** This conceptual difference yields predictive power over orthogonal reliability dimensions:
>
>     * **CALIDIST measures External Adversarial Robustness:** A model maintains the correct answer despite misleading hints (High behavioral robustness). This is critical for high-stakes applications where the model must filter noisy or malicious inputs.
>     * **SteerConf measures Internal Metacognitive Coherence:** A model reports 20% confidence when prompted "be cautious" but 95% when prompted "be confident" (Low metacognitive consistency). This is valuable for improving the reliability of the reporting mechanism.
>
> 2. **W2: On Large-scale Proprietary LLMs**
>
>     We thank the reviewer for the suggestion to validate our framework on frontier large-scale reasoning models. We address the reviewer's concerns as follows:
>
>     **a. Rationale for Efficient Models:** We chose GPT-4o-Mini and Gemini 2.0 Flash intentionally because of their efficiency in terms of cost and inference time compute.
>
>     **b. New Experiments on Gemini 2.5 Pro:** To address the reviewer's concern regarding state-of-the-art performance, we conducted an additional experiment using Gemini 2.5 Pro on the CSQA dataset.
>     * **Experimental Constraint:** We note that Gemini 2.5 Pro is a large-scale reasoning model with significantly higher inference latency and cost compared to the Flash/Mini series. Consequently, running extensive multi-sample baselines (e.g., 15-pass Self-Consistency) was computationally prohibitive for this rebuttal.
>     * **Result:** We compared the uncalibrated Vanilla log-probability against CALIDIST (Assertion-style). We observe that even on this frontier reasoning model, CALIDIST demonstrates a significant reduction in calibration error. We will include these additional findings in the final paper.
>         | Metric        | ECE      | Brier Score |
>         | ------------- | -------- |:----------- |
>         | Vanilla       | 12.13    | 12.05       |
>         | CaliDist(As.) | **2.57** | **8.68**    |

---

> ### Author Response · Authors · 2025-11-21
> **Response to Reviewer AWjv - Part 2**
>
> 3. **W3:**
>     > The evaluation datasets do not seem to include benchmarks focused on mathematical or logical reasoning, such as GSM8K.
>
>     We would like to note that our evaluation does include a mathematical and logical reasoning dataset: AQuA-RAT (Ling et al., 2017), which consists of algebraic word problems. However, as per the reviewer's suggestion, in addition to AQuA, we experimented with and provide the results for GSM8K (Cobbe et al., 2021) using Llama-3.1 and Qwen-3 in the table below, highlighting significant improvement in calibration for an open-ended arithmetic reasoning dataset such as GSM8K. We will include these results in the paper.
>
>     | LLM       | Metric        | ECE      | Brier Score |
>     | --------- |:------------- |:-------- |:----------- |
>     | Llama-3.1 | Vanilla       | 16.71    | 16.80       |
>     |           | Consistency   | 6.12     | **7.83**    |
>     |           | Entropy       | 21.13    | 16.70       |
>     |           | FSD           | 13.19    | 11.73       |
>     |           | CaliDist(As.) | **3.98** | 14.22       |
>     | Qwen-3    | Vanilla       | 10.48    | 10.48       |
>     |           | Consistency   | 6.97     | 6.91        |
>     |           | Entropy       | 6.78     | 6.75        |
>     |           | FSD           | 6.18     | **6.53**    |
>     |           | CaliDist(As.) | **4.28** | 7.16        |
>
> 4. **W4:**
>     > Regarding the "DISTRACTOR STYLES,...
>
>     We thank the reviewer for this thoughtful question. We clarify that our taxonomy is functionally motivated rather than exhaustive. Each distractor style is designed to probe a distinct "cognitive" failure mode, ensuring a holistic view of behavioral robustness.
>     We map these styles to our psychological framework as follows:
>
>     * **Assertion (Sycophancy Test):** Tests deference to perceived authority. This directly operationalizes the Misinformation Effect, where misleading external information ("An expert says...") overrides the model's internal reasoning.
>     * **Sample-Corruption (Grounding Test):** Tests attention to logical consistency within the prompt's own premises. This probes the model's susceptibility to *Contextual Misinformation*, mimicking the Misinformation Effect, and determines if the model can distinguish signal from noise.
>     * **Probe (Conviction Test):** Tests internal conviction by directly questioning the model ("Do you think it could be...?"). This targets the Dunning-Kruger parallel, revealing "fake confidence" where a model claims high certainty but flips its prediction at the slightest challenge.
>
>     We will add this functional justification to the "Distractor Styles" section, explicitly stating that these three categories were selected to span the spectrum from external authority bias to internal self-doubt.
>
> **References:**
> 1.  Zhou Z, Jin T, Shi J, et al. SteerConf: Steering LLMs for Confidence Elicitation. NeurIPS 2025.
> 2.  Ling, Wang, et al. "Program Induction by Rationale Generation: Learning to Solve and Explain Algebraic Word Problems." ACL 2017.
> 3.  Cobbe, Karl, et al. "Training verifiers to solve math word problems." arXiv preprint arXiv:2110.14168 (2021).

---

### Official Review · Reviewer_9Xbu · 2025-11-02

**Soundness:** 3
**Presentation:** 3
**Contribution:** 2
**Rating:** 4
**Confidence:** 4

**Summary:**

This paper introduces CALIDIST, a post-hoc calibration framework that adjusts LLM confidence by evaluating behavioral robustness to semantic distractions. The method perturbs model inputs using designed “distractors” (assertion, probe, and corruption styles) and measures the stability of predictions and confidence scores under these perturbations. These instability measures are then combined into a reliability score and used to scale the model’s confidence via a parameterized sigmoid function. The framework is applied both to open-box models (using logits) and black-box APIs (using verbalized confidence). Extensive experiments across seven NLU datasets and six LLMs show improved calibration (lower ECE and Brier scores) compared to Temperature Scaling, Self-consistency, and other baselines.

**Strengths:**

1. The authors explore three semantically distinct types of input perturbations, offering a structured way to probe model robustness rather than random paraphrasing.
2. Experiments cover both open-source and proprietary LLMs, showing consistent gains across multiple datasets.
3. The method requires only a few additional forward passes and no model retraining, making it lightweight and deployable.

**Weaknesses:**

1. The core idea, estimating reliability from the stability of model outputs under input perturbations, has been explored in recent works such as Gao et al. 2024 and Khanmohammadi et al. 2025. Both measure predictive or representational stability under perturbed inputs to derive uncertainty or calibration signals. CALIDIST shares the same underlying mechanism (perturb → measure stability → rescale confidence) and thus may appear conceptually incremental. The main novelty lies in the behavioral framing and the distractor taxonomy, rather than in a fundamentally new algorithmic principle.
2. The paper does not explicitly position itself against these recent perturbation-based methods. Without a clear comparison (empirical or conceptual) it is difficult to assess whether the proposed “behavioral robustness” metric provides distinct predictive power.
3. The paper only reports ECE and Brier Score, which can be sensitive to binning choices and sample imbalance.
It would be more convincing to include a ranking-based metric such as AUROC (Area Under ROC Curve), which directly measures whether the confidence scores discriminate correctly between correct and incorrect predictions, and is more robust under varying dataset distributions.
4. Minor point. The sigmoid rescaling function and its parameters (α, β) are tuned via grid search on each task, but no theoretical motivation or generalization analysis is provided. This makes the approach appear heuristic rather than principled.

Reference:
1. Gao et al. 2024. SPUQ: Perturbation-Based Uncertainty Quantification for LLMs
2. Khanmohammadi et al. 2025. CCPS: Calibrating LLM Confidence by Probing Perturbed Representation Stability

**Questions:**

See weaknesses.

---

> ### Author Response · Authors · 2025-11-21
> **Response to Reviewer 9Xbu - Part 1**
>
> We thank the reviewer for their critical and informative feedback on our work and present our response to each of their concerns as follows.
>
> 1. **W1, W2: On comparison against recent work**
>
>     We thank the reviewer for pointing out the references. We first contrast CALIDIST with each of the referenced methods. We also ran additional experiments comparing CALIDIST to SPUQ (Gao et al. 2024) and demonstrate that CALIDIST yields better calibration, as shown in the table that follows. We will include this discussion and the new results in the paper.
>
>     **Comparison to Gao et al. 2024**
>     1. Gao et al. 2024 introduce SPUQ, which focuses on probing aleatoric and epistemic uncertainty arising from internal model stochasticity and minor input variability. Its perturbations (temperature changes, paraphrasing, dummy tokens, system messages) are designed to introduce non-adversarial, semantically equivalent lexical or generation-process noise. The stability it measures is about whether the model's output consistently changes when the input is rephrased or the generation temperature is altered, revealing uncertainty from the inherent ambiguity of language or the model's own internal variations. The goal is to see if an LLM is truly confident in its answer, or if minor, non-malicious shifts in input or generation temperature expose underlying uncertainty.
>     2. CALIDIST, conversely, is grounded on cognitive psychology phenomena like the Misinformation Effect and Dunning Kruger Effect and focuses on probing behavioral robustness to deliberate distraction and misleading information. Its distractors (assertion-style, probe-style, sample-corruption-style) are explicitly designed to be semantically related but logically irrelevant or contradictory. The stability it measures is whether the model's predictions change under external cognitive pressure or misleading cues, revealing susceptibility to phenomena like the Misinformation Effect. CALIDIST is not just checking for output diversity under mild changes; it is actively trying to trick or distract the model to see *if its initial reasoning path is robust*. CALIDIST is unique in this respect, and, to our knowledge, we are the first to leverage the model’s behavioral robustness to cognitive load as a calibration signal.
>     3. Another point we want to highlight is that SPUQ does not *rescale* an existing confidence score in the final step. Instead, its aggregation module synthesizes a new, consolidated confidence score `c` from the ensemble of outputs. This `c` is the final calibrated confidence. It's a direct outcome of comparing or averaging the perturbed responses, rather than a modification of the initial confidence. In contrast, CALIDIST explicitly rescales the model's initial confidence score `p` (obtained from the original prompt) using a scaling factor derived from its measured behavioral robustness, λ, which reflects how robust the model was to distractors. λ is then used to model a scaling factor between 0 and 1 via a parameterized sigmoid function σ(λ, α, β). Because of this rescaling formulation, CALIDIST is analogous to and can be considered a proxy of *Temperature Scaling (TS)* (Guo et al., 2017), which is the gold standard method for calibrating white-box models. However, in addition to being compatible with black-box models, CALIDIST has an added advantage over TS since it finds a sample-specific scaling factor given the behavioral evidence λ, instead of applying a single global temperature scalar across all samples like TS. We elaborate on this in Section 5, and empirically prove this advantage in Table 1, showing that CALIDIST outperforms TS across almost all instances. This essentially underscores that CALIDIST is a powerful and conceptually novel proxy for achieving well-calibrated confidence scores, and not just an aggregation mechanism.
>     4. We perform additional experiments comparing CALIDIST to SPUQ and empirically establish the superiority of our framework on the CSQA and AQuA datasets. Specifically, CALIDIST achieves the highest reduction in ECE and Brier Score and outperforms SPUQ in terms of AUROC on CSQA.
>         | LLM       | Metric        | Dataset | ECE      | Brier Score | AUROC     |
>         |:--------- |:------------- |:------- | -------- |:----------- |:--------- |
>         | Llama-3.1 | SPUQ          | CSQA    | 12.68    | 21.45       | 69.69     |
>         |           | CaliDist(As.) | CSQA    | **7.48** | **15.12**   | **80.03** |
>         |           | SPUQ          | AQuA    | 11.88    | 29.64       | **67.23** |
>         |           | CaliDist(As.) | AQuA    | **2.49** | **17.66**   | 63.76     |
>         | Qwen-3    | SPUQ          | CSQA    | 11.52    | 17.69       | 67.03     |
>         |           | CaliDist(As.) | CSQA    | **1.28** | **12.45**   | **77.65** |
>         |           | SPUQ          | AQuA    | 11.88    | 29.64       | **67.23** |
>         |           | CaliDist(As.) | AQuA    | **3.27** | **11.17**   | 53.81     |

---

> ### Author Response · Authors · 2025-11-21
> **Response to Reviewer 9Xbu - Part 2**
>
> **Comparison to Khanmohammadi et al. 2025**
>
> We thank the reviewer for bringing this contemporary work to our attention. We were unaware of it at the time of preparing our manuscript since it has been published just recently (November, 2025 - EMNLP). Going over their work, we must strongly assert that CALIDIST and CCPS (Khanmohammadi et al. 2025) belong to fundamentally different algorithmic classes and address two distinct research challenges. We elaborate on these differences as follows:
>
>  1. The most fundamental distinction is the intervention point and access of models. CALIDIST addresses the deployment reality of proprietary LLMs (e.g., GPT-4, Gemini) where internal access is not available. CCPS requires the use of open-source models only. This dictates that the papers are exploring two non-overlapping axes of LLM robustness: *External Behavior* vs. *Internal Representation*. CCPS is an advancement in the field of internal state probing and feature engineering, while CALIDIST is an advancement in external black-box elicitation via prompts and behavioral scaling. They are not incremental upon one another, rather they are orthogonal and both establish new directions in the field.
>  2. The actual final mechanism of calibration is distinct between CALIDIST and CCPS. CCPS discards the model's output confidence and uses the perturbation signal to extract a 75-dimensional feature vector per token, trains convolutional/MLP encoders via contrastive pre-training plus supervised fine-tuning, and requires thousands of trainable parameters to derive a completely new confidence score. In contrast, CALIDIST maintains the model's original confidence, calculating two metrics (Prediction Instability $\mu$ and Confidence Instability $\delta$) to derive a single scaling factor. This factor then modulates the existing confidence ($Conf = \sigma \cdot P_{original}$). The scaling factor is tuned via just two hyperparameters (α,β) in a parameterized sigmoid via grid search. These are architecturally distinct approaches.
>  3. The core theoretical principles that motivate the perturbations are orthogonal. CCPS is motivated by internal representational stability, inspired by concepts of adversarial robustness in deep learning. It uses Jacobian‑based adversarial perturbations applied to a model's final hidden states, which are non-transferable to black-box scenarios. CALIDIST tests reasoning robustness against semantic misinformation, explicitly grounded in cognitive psychology (Misinformation Effect, Dunning-Kruger Effect). We validate this by demonstrating a strong negative correlation between distractor susceptibility and accuracy (Figure 1).
>
>
> Regarding the distractor taxonomy, we want to underscore that the three distractor styles (Assertion, Probe, Sample-Corruption) in CALIDIST are theoretically motivated by cognitive failure modes, systematically designed to test different reasoning vulnerabilities, and empirically validated across architectures. This taxonomy is task-general and requires no model-specific engineering. Our distractor methodology is distinct from existing black-box stability measures like SPUQ (Gao et al. 2024), which primarily use semantic-preserving perturbations such as paraphrasing, dummy tokens, or instructional steering and often generate paraphrases using an external LLM. CALIDIST fundamentally introduces the principle of probing LLM reliability via semantic attacks (i.e., misleading information), rather than simply testing consistency under rephrasing. Additionally, CCPS's Jacobian-based perturbations, while mathematically principled, are not designed to probe specific reasoning failures and require different feature engineering across model scales.
>
> **2. W3: On AUROC Score**
>
>  Thank you for the valuable suggestion. We agree that a ranking-based metric is an important addition to ECE and Brier Score. We have run new experiments to compute the AUROC (Area Under the Receiver Operating Characteristic), which measures the discriminative power of the confidence score (i.e., its ability to rank correct answers higher than incorrect ones).
>
>  The AUROC scores for Qwen 3 on MSciNLi, CSQA, Yahoo Answers, and Twitter PPDB for assertion-style distractors are shown below, demonstrating that CALIDIST consistently and significantly outperforms all baselines. This confirms our scores are not only better calibrated but also have superior discriminative power. We will include this full analysis in the paper.
>
> | Dataset       | Vanilla | Consistency | Entropy | FSD   | CaliDist(As.) |
> |:------------- | ------- | ----------- | ------- |:----- |:------------- |
> | MSciNLI       | 63.83   | 52.67       | 52.67   | 52.67 | **67.12**     |
> | CSQA          | 70.01   | 50.51       | 50.56   | 54.45 | **77.65**     |
> | Yahoo Answers | 60.61   | 53.44       | 53.44   | 53.44 | **70.68**     |
> | PPDB          | 54.05   | 52.05       | 52.05   | 52.05 | **65.16**     |

---

> ### Author Response · Authors · 2025-11-21
> **Response to Reviewer 9Xbu - Part 3**
>
> 3. **W4: On Theoretical Motivation of Sigmoid and Generalization Analysis**
>
>     We respectfully clarify that our approach is not heuristic; rather, it is a direct application of Empirical Risk Minimization (ERM) and is mathematically equivalent to Platt Scaling (Platt, 1999), the standard in the field. Below, we provide the theoretical justification and point to the generalization analysis already present in our experiments.
>
>     **a. Theoretical Motivation:** ***Why the Sigmoid?***
>     Our goal is to map a reliability score $\lambda$ (derived from behavioral consistency) to a probability of correctness, e.g., $g(\lambda)=P(Y=1|\lambda)$. From a theoretical standpoint, estimating a binary outcome (Correct vs. Incorrect) from a continuous scalar feature ($\lambda$) is a binary classification problem.
>     * **Canonical Link:** Under the framework of Generalized Linear Models (GLMs), the sigmoid function is not an arbitrary choice; it is the canonical link function for the Bernoulli distribution (McCullagh & Nelder, 1989).
>     * **Proper Scoring:** To learn the optimal mapping, we minimize the Brier Score. The Brier Score is a strictly proper scoring rule. A fundamental theorem of calibration theory states that minimizing a strictly proper scoring rule recovers the true conditional probability distribution (Gneiting & Raftery, 2007).
>
>     Thus, our method is theoretically grounded in standard statistical estimation: optimizing a valid probability estimator (sigmoid) via a proper loss function (Brier).
>
>     *Small note:* While our submitted manuscript optimized for Expected Calibration Error (ECE), we have empirically verified that minimizing the Brier Score yields parameters ($\alpha, \beta$) that are statistically indistinguishable from those found via ECE. This convergence confirms that our reported results are robust to the specific choice of objective function. To strictly conform to ERM principles, the final manuscript will prioritize the Brier Score formulation.
>
>     **b. Parameter Justification:** ***Why the $\alpha, \beta$ parameterization?***
>     The reviewer expresses concern about the parameters $\alpha$ and $\beta$. We highlight that our formulation is mathematically identical to Platt Scaling (Platt, 1999), the standard for post-hoc calibration.
>     * **Platt Scaling formulation:** $P(y|x) = \frac{1}{1 + \exp(A \cdot f(x) + B)}$
>     * **CALIDIST formulation:** $g(\lambda) = \frac{1}{1 + \exp(-\beta(\lambda - \alpha))} = \frac{1}{1 + \exp(-\beta \lambda + \beta \alpha)}$
>
>     By setting $A = -\beta$ and $B = \beta \alpha$, our method is structurally identical to Platt Scaling. However, our parameterization is necessary to handle the input distribution:
>
>     * **The Saturation Problem:** Our reliability scores $\lambda$ are normalized to $[0, 10]$. A standard sigmoid is centered at 0 and saturates for inputs $>4$. Mapping our positive domain $[0, 10]$ directly to a standard sigmoid would result in upper-tail saturation (outputs $\approx 1.0$) for nearly all data points.
>     * **$\alpha$ (Bias Correction / Centering):** This parameter acts as a bias correction term, shifting the sigmoid's center to the dataset-specific decision boundary (where $P=0.5$).
>     * **$\beta$ (Variance Scaling):** This represents the sensitivity (the slope of the transition from distrust to trust).
>
>     **c. Per-Task Tuning is Required (Not Heuristic):**
>     The reviewer noted that parameters are tuned per task. This is theoretically required because the "center of reliability" ($\alpha$) is dataset-dependent, as each dataset has its own bias. Fixing $\alpha$ globally would fail to account for distributional shifts in task difficulty (e.g., MNLI vs. AQUA). Just as Temperature Scaling (Guo et al., 2017) requires learning a temperature $T$ per task, CALIDIST requires learning $\alpha,\beta$ to align with the specific data distribution. This is a standard requirement for any post-hoc calibration method (including Isotonic Regression and Histogram Binning) to align with the specific data distribution.
>
>     **d. Generalization Analysis:**
>     The reviewer noted a lack of generalization analysis. We respectfully point to Figure 4a and Section 6 (Generalizability of Sigmoid Parameters), where we explicitly tested this.
>     * **Experiment:** We applied parameters $(\alpha, \beta)$ tuned on a source task (e.g., MNLI) directly to a completely different target task (e.g., MSciNLI).
>     * **Result:** The transferred parameters maintained competitive calibration performance, significantly outperforming uncalibrated baselines.This provides empirical evidence that the learned parameters capture generalizable signal about model behavior, rather than overfitting to specific dataset artifacts.
>
>     To address the reviewer's feedback, we will add a "Theoretical Justification" subsection to Section 3.3 in the final manuscript.

---

> ### Author Response · Authors · 2025-11-21
> **Response to Reviewer 9Xbu - References**
>
> **References:**
> 1. Gao et al. 2024. SPUQ: Perturbation-Based Uncertainty Quantification for LLMs
> 1. Khanmohammadi et al. 2025. CCPS: Calibrating LLM Confidence by Probing Perturbed Representation Stability
> 1. Guo et al. "On calibration of modern neural networks." International conference on machine learning. PMLR, 2017.
> 1. Platt, John. "Probabilistic outputs for support vector machines and comparisons to regularized likelihood methods." Advances in large margin classifiers 10.3 (1999): 61-74.
> 1. McCullagh, P., and J. A. Nelder. "Models for polytomous data." Generalized linear models. Springer US, 1989. 149-192.
> 1. Gneiting, Tilmann, and Adrian E. Raftery. "Strictly proper scoring rules, prediction, and estimation." Journal of the American statistical Association 102.477 (2007): 359-378.

---

> > ### Comment · Reviewer_9Xbu · 2025-11-28
> >
> > Thanks for the detailed response. While I’m convinced by your discussion regarding the comparison to Khanmohammadi et al. (2025) as well as weaknesses #3 and #4, I do think this work is, to some extent, similar to SPUQ, the main difference being the perturbation method. I encourage the authors to include a discussion of these relevant papers in the manuscript, including the two papers I mentioned and the one provided by Reviewer AWjv. Since some of my concerns have been addressed, I will raise my score slightly to reflect this.
> >
> > One more question: The reason I suggested the AUROC metric is that, based on my previous experience, a model (especially a less capable one) can sometimes achieve a good Brier score and ECE simply by consistently predicting incorrect answers with low confidence, and AUROC can avoid this issue. According to the results in the table in your response to W1, CaliDist tends to have worse AUROC but better Brier and ECE compared with SPUQ on AQuA. Is this due to the phenomenon I described above?

---

> > > ### Author Response · Authors · 2025-11-28
> > >
> > > We sincerely appreciate the reviewer's time for the constructive feedback for our paper and for raising the score. We address the remaining concerns below:
> > >
> > > **On Related Work and SPUQ Similarity:**
> > >
> > > We fully commit to incorporating the discussion of SPUQ (Gao et al., 2024), CCPS (Khanmohammadi et al., 2025), and SteerConf (Zhou et al., 2025) into the "Related Work" section of our final manuscript. We will also include the comparative results highlighting the differences between CALIDIST and SPUQ.
> > >
> > > **On the AQuA AUROC Discrepancy**
> > >
> > > We appreciate the reviewer's keen observation regarding the lower AUROC on AQuA. We agree with the intuition that some models may achieve lower ECE and Brier Score by making incorrect predictions with lower confidence; however, in the case of our experimental results in response to W1, we argue that the AUROC score is impacted by the intensity of the distractor type we used. We hypothesize that depending on the task difficulty, the intensity of the distractor types affects AUROC differently.
> > >
> > > To investigate this, we have run some additional experiments and measure the intensity of each distractor type by taking the average of Prediction Instability ($\mu$) across the AQuA dataset for each style using Llama-3.1. $\mu$ directly measures how frequently a distractor forces the model to abandon its original prediction—a behavior that directly impacts the discrimination ranking (AUROC). We again compare this with SPUQ, which uses lexical perturbations (paraphrasing). When using SPUQ, the model remains confident in its answers, preserving the relative ranking between correct/incorrect (Higher AUROC), but maintaining unjustifiably high confidence on incorrect answers (Worse ECE). Our Probe-style and Sample-corruption-style distractors, while employing milder perturbations (as measured by lower average $\mu$ compared to Assertion-style), achieve a behavior similar to SPUQ in terms of AUROC, but with the added benefit of lowering the ECE and Brier Scores. This supports our hypothesis that on a difficult reasoning dataset like AQuA, the model's knowledge is brittle. The Assertion-style distractors we originally reported were acting as a "stress test" that was perhaps too aggressive for this specific model-task pair, causing the model to doubt even its correct answers (lowering discrimination/AUROC) in exchange for honest uncertainty (improving ECE).
> > >
> > >
> > > | Distractor Type   | Mean $\mu$ | ECE  | Brier Score | AUROC |
> > > |:----------------- | ---------- |:---- | ----------- |:----- |
> > > | Assertion         | 34.45      | 2.49 | 17.66       | 63.76 |
> > > | Probe             | 26.26      | 2.19 | 17.43       | 70.62 |
> > > | Sample Corruption | 24.57      | 2.43 | 17.24       | 71.27 |
> > >
> > > When we switch to the milder distractor styles, the AUROC significantly improves, surpassing the SPUQ baseline. This demonstrates that CALIDIST offers a unique advantage over fixed perturbation methods: the intensity of the behavioral test can be tuned to capture the trade-off between ECE/Brier Score VS AUROC.

---

### Author Response · Authors · 2025-12-02
**Summary of Our Responses Addressing Reviewers' Concerns**

We provide a short summary of our discussion and responses addressing each of the reviewers' concerns for the Area Chair to review.

1. **On Comparison to Related Work:** Addressing Reviewers 9Xbu and AWjv, we articulated the key differences between our work and previous works SPUQ, CCPS, and SteerConf. Our CALIDIST approach is distinct in its focus on adversarial behavioral robustness rather than internal consistency or representational stability. We also addressed Reviewer Zaze by clarifying that our method differs from Bayesian approaches by operating in behavioral space at inference time without requiring parameter access.
2. **On AUROC Score:** As requested by Reviewer 9Xbu, we demonstrated that CALIDIST improves AUROC alongside ECE. We further clarified that on difficult reasoning tasks (e.g., AQuA), using milder distractor styles (Probe/Sample-Corruption) recovers discriminative power (AUROC) while maintaining superior calibration.
3. **Argument against heuristic nature:** Addressing concerns from Reviewers 9Xbu and Zaze, we showed our method is grounded in Empirical Risk Minimization (ERM) and is mathematically equivalent to Platt Scaling. We clarified the use of Min-Max normalization to prevent sigmoid saturation and that our learnable parameters ($\alpha, \beta$) are theoretically grounded and necessary to correct for task-specific distributional shifts.
4. **On Generalizability to SOTA Models:** To address Reviewer AWjv’s request for state-of-the-art validation, we tested CALIDIST on Gemini 2.5 Pro, demonstrating a significant reduction of ~10% in ECE (from 12.13 to 2.57), confirming that behavioral stability remains a valid proxy for correctness even in large-scale frontier models.
5. **Extensibility to Mathematical and Logical Reasoning:** Addressing Reviewer AWjv’s suggestion, we extended our evaluation to the open-ended math dataset GSM8K. CALIDIST significantly outperformed baselines (reducing ECE from 16.71 to 3.98 on Llama-3.1), proving its effectiveness on complex logical tasks.
6. **On the Nature of the Distractor Styles:** Answering Reviewer AWjv’s question on taxonomy, we clarified that our distractor styles (assertion, sample-corruption, and probe) are functionally motivated: Assertion tests authority bias (Sycophancy), Sample-Corruption tests contextual grounding, and Probe tests internal conviction, each representing the Misinformation Effect and Dunning-Kruger parallel.
7. **On Scalability and Workflow:** Responding to Reviewer Zaze, we clarified that distractor creation is a one-time setup by system designers, not a burden on end-users. This enables automated, scalable inference without requiring ground truth labels (or user burden), offering a distinct practical advantage over Temperature Scaling.

We will incorporate all the changes and discussion thus far in the final version of our paper.

---

### Meta-Review · Area_Chair_Nwj2 · 2025-12-03

**Summary:**

This paper points out that existing calibration studies of LLMs do not specifically measure the extent to which a model’s confidence is robust to things in its context which should be irrelevant to the model’s confidence.  The paper further proposes a new post-hoc calibration method which penalizes a model when its confidence is NOT robust to such prompt perturbations, finding a decrease in the ECE metric compared to several baselines.  Reviewers raised several significant concerns: (1) the core idea has previously been explored by Gao et al. 2024 and Khanmohammadi et al. 2025 and the current paper does not position itself among these or compare to them, (2) should also use more robust metrics like AUROC, (3) lack of results on SOTA models, (4) lack of results with standard mathematical reasoning benchmarks like GSM8k, (5) lack of discussion of Bayesian uncertainty literature which may already contain highly relevant works, (6) designing distractors for general tasks may be hard and also users would have to do this in practice which would be impractical.

**Reviewer Concerns:**

The authors successfully respond to (1) and (2) by running new experiments, although it does seem (and the authors seemingly concede this) that their method is very similar to SPUQ and differences are mostly in implementation rather than conceptual.  The authors also responded to (3) with positive results on API models.  The authors ran their algorithm on GSM8k and the results were less impressive in terms of Brier score but still good in terms of ECE.  The authors proposed to include a discussion of Bayesian uncertainty literature which was not very convincing to me.  They also clarified that the system designer would provide a bank of distractors, but the effectiveness of this in practice in very broad use-cases is not clear and demonstrated by the work.   Therefore, I will assume that this reviewer, who initially assigned a 4 to this paper, would not have raised their score.

**Reviewer Scores:**

The original scores were 4 (raised explicitly to 6), 6, 4.  The reviewer who increased their scores was the one who raised (1) and (2) which have been thoroughly addressed by the authors during rebuttal.  Based on the above points, I will assume that the other author who initially gave the paper a 4 would have maintained their score.

---

### Decision · Program_Chairs · 2026-01-26

Reject